

# A multi-species data assimilation system to retrieve information on land-atmosphere exchange processes

Ivar R. van der Velde[1,2,3], John B. Miller[2], Michiel K. van der Molen[1], Pieter P. Tans[2], Bruce H. Vaughn[4], James W.C. White[4], Kevin Schaefer[5], and Wouter Peters[1,6]

[1]Department of Meteorology and Air Quality, Wageningen University and Research, Wageningen, The Netherlands
[2]Global Monitoring Division, NOAA Earth System Research Laboratory, Boulder, CO, United States
[3]Cooperative Institute for Research in Environmental Sciences, University of Colorado, Boulder, CO, United States
[4]Institute for Arctic and Alpine Research, University of Colorado, Boulder, CO, United States
[5]National Snow and Ice Data Center, University of Colorado, Boulder, CO, United States
[6]Centre for Isotope Research, University of Groningen, Groningen, The Netherlands

*Correspondence to:* Ivar van der Velde (Ivar.vanderVelde@noaa.gov)

**Abstract.** To improve our understanding of the global carbon balance and its representation in terrestrial biosphere models we present here a first multi-species application of the CarbonTracker Data Assimilation System (CTDAS). The system's modular design allows for assimilating multiple atmospheric trace gases simultaneously to infer exchange fluxes at the Earth surface. In the prototype discussed here we interpret signals recorded in observed carbon dioxide ($CO_2$) along with observed ratios of its

stable isotopologues $^{13}CO_2/^{12}CO_2$ ($\delta^{13}C$). The latter is in particular a valuable tracer to untangle $CO_2$ exchange from land and oceans. Potentially, it can also be used as a proxy for continent-wide drought stress in plants, largely because the ratio of $^{13}CO_2$ and $^{12}CO_2$ molecules removed from the atmosphere by plants is dependent on moisture conditions.

The multi-species CTDAS system varies the net exchange fluxes of both $^{13}CO_2$ and $CO_2$ in ocean and terrestrial biosphere models to create an ensemble of $^{13}CO_2$ and $CO_2$ fluxes that propagates through an atmospheric transport model. Based on

differences between observed and simulated $^{13}CO_2$ and $CO_2$ mole fractions (and thus $\delta^{13}C$) our Bayesian minimization approach solves for weekly adjustments to both net fluxes and isotopic terrestrial discrimination that minimizes the difference between observed and estimated mole fractions.

With this system we are able to estimate changes in terrestrial $\delta^{13}C$ exchange on seasonal and continental scales in the Northern hemisphere where the observational network is most dense. Our results indicate a decrease in stomatal conductance

on a continent-wide scale during a severe drought. These changes could only be detected after applying combined atmospheric $CO_2$ and $\delta^{13}C$ constraints as done in this work. The additional constraints on surface $CO_2$ exchange from $\delta^{13}C$ observations neither affected the estimated carbon fluxes, nor compromised our ability to match observed $CO_2$ variations. The prototype presented here can be of great benefit not only to study the global carbon balance but potentially also to function as a data driven diagnostic to assess multiple leaf-level exchange parameterizations in carbon-climate models that influence the $CO_2$,

water, isotope, and energy balance.



## 1 Introduction

The terrestrial biosphere has absorbed about 25 % of global fossil fuel carbon dioxide ($CO_2$) emissions over the last several decades but the future of this sink is highly uncertain in a warming world (*Booth et al.*, 2012; *Rowlands et al.*, 2012). It depends on the small difference between two large fluxes of the terrestrial carbon cycle: photosynthetic uptake or gross primary

production (GPP) and terrestrial ecosystem respiration (TER), and is here referred to as the net ecosystem exchange (NEE = TER − GPP + fire disturbances and land use change and harvesting of crops). All these flux terms respond to changes in local temperature, precipitation, nutrient availability and other key environmental variables (*Friedlingstein et al.*, 2006). Extreme climate events such as droughts can decrease GPP and increase TER and fire disturbances to a point where regional NEE is turned into a temporary carbon source (*Ciais et al.*, 2005; *Gatti et al.*, 2014; *Van der Laan-Luijkx et al.*, 2015). These dynamic

responses (and positive feedbacks whereby increased $CO_2$ may lead to more droughts) are now an integral part of climate models that include fully coupled carbon cycling (*Booth et al.*, 2012; *Dai et al.*, 2012). Such models give rise to a wide range of climate projections primarily as a result of different simulations of terrestrial carbon exchange (*Friedlingstein et al.*, 2006). It is therefore important to test and improve the representation of the terrestrial biosphere in carbon-climate models. Uncertainty in climate projections can be reduced by evaluating present day performance of these models to observations (*Hoffman et al.*,

2014). This paper presents a data assimilation system that can be used to evaluate existing terrestrial biosphere models by using an extensive number of atmospheric $CO_2$ observations in tandem with other trace gases.

Measurements of atmospheric $CO_2$ have been used to infer carbon fluxes at the Earth's surface using a variety of inversion techniques (e.g., *Keeling and Revelle*, 1985; *Keeling et al.*, 1989; *Tans et al.*, 1993; *Ciais et al.*, 1995; *Rayner et al.*, 2008; *Alden et al.*, 2010). Unfortunately, a limited number of $CO_2$ observations, errors in atmospheric transport modeling, and the

realism of bottom-up carbon flux estimates are limiting the utility of these techniques. For instance, the representation of subgrid scale vertical motion in (and through the top of) the planetary boundary layer is one of the most uncertain aspects in atmospheric tracer modeling and can hinder the accuracy of $CO_2$ transport (*Kretschmer et al.*, 2012; *Miller et al.*, 2015). In addition, atmospheric $CO_2$ as a tracer has its own limitations as it only reflects the a small residual of different sources and sinks, such as wild fires, anthropogenic sources, ocean in- and outgassing, and terrestrial GPP and TER.

The CarbonTracker Data Assimilation System (CTDAS) has been developed to estimate global net ocean and terrestrial carbon exchange fluxes, with a focus on North America and Europe (*Peters et al.*, 2005, 2007, 2010; *Van der Laan-Luijkx et al., in prep.*, 2017). This application uses the Ensemble Kalman Filter (EnKF) as a Bayesian minimization approach for the estimation of weekly ocean and terrestrial carbon fluxes on a $1 \times 1$ degree horizontal grid to improve the agreement between modeled and measured atmospheric $CO_2$. The versatile object-oriented design of CTDAS allows flexible implementation of

different components of the data assimilation system (*Van der Laan-Luijkx et al., in prep.*, 2017). Such modifications include but are not limited to, (1) the configuration of the state vector, (2) the expansion of the monitoring network, such as for the Amazon (*Van der Laan-Luijkx et al.*, 2015) and China (*Zhang et al.*, 2014), (3) the use of Lagrangian atmospheric transport (*He et al., in prep.*, 2017), and (4) to monitor other tracer gases like methane (*Bruhwiler et al.*, 2014; *Tsuruta et al.*, 2016).



One aspect that has not yet been explored in CTDAS is the monitoring of multiple trace gases in the atmosphere that are strongly related (i.e., gases with a common chemical or metabolic pathway in the ocean and/or terrestrial biosphere). The main purpose of such an application is to improve the estimation of carbon fluxes and to retrieve new information on the underlying flux exchange processes that would otherwise remain undetected. We are in particular interested in the use of the stable isotope $^{13}C$ (in atmospheric $CO_2$) as an additional tracer alongside total $CO_2$ to estimate carbon sources and sinks and their variability. In earlier studies $^{13}C$ was used to distinguish oceanic from terrestrial carbon exchange, as oceans take up $^{13}CO_2$ more efficiently than land surfaces relative to $^{12}CO_2$. In so-called double-deconvolution methods this particular trait is used to untangle the global land carbon budget from ocean carbon budget (*Keeling et al.*, 1989; *Tans et al.*, 1993; *Ciais et al.*, 1995). More recently $^{13}C$ isotope was used to study the diurnal cycle of GPP and TER (*Wehr et al.*, 2016) and was used as a tracer of water use efficiency to study long-term responses to $CO_2$ increases in tree-rings (*Van der Sleen et al.*, 2015), and attempts are underway to do the same based on atmospheric records. On regional scales variations in the ratio of $^{13}CO_2/^{12}CO_2$ (typically reported as $\delta^{13}C$ in ‰ relative to the VPDB reference ratio) reflect changes in discrimination processes associated with photosynthetic uptake of carbon by plants (e.g., *Farquhar et al.*, 1989; *Fung et al.*, 1997; *Scholze et al.*, 2003; *Rayner et al.*, 2008). Plants generally take up the heavier $^{13}CO_2$ molecules less efficiently than $^{12}CO_2$ molecules, increasing the $^{13}CO_2/^{12}CO_2$ ratio of $CO_2$ remaining in the atmosphere. This kind of discrimination against $^{13}C$ is much stronger for $C_3$ plants than for $C_4$ plants, but also varies as a function of moisture conditions in the canopy air and soil (*Farquhar et al.*, 1980, 1989; *Ekblad and Högberg*, 2001; *Ometto et al.*, 2002; *Suits et al.*, 2005). That implies that under the right circumstances, measured atmospheric $\delta^{13}C$ can be used to recognize land usage, such as $C_3/C_4$ photosynthesis, and changes in photosynthetic activity resulting from droughts stress (*Ballantyne et al.*, 2010; *Raczka et al.*, 2016).

Such an application could also be beneficial to explore other facets of carbon exchange. Any errors in the fossil fuel emission inventories (although relatively small) are in the current CTDAS releases aliased erroneously on the natural ocean and terrestrial fluxes. Assimilation of the fraction of the radioactive isotope $^{14}CO_2$ in the atmosphere would allow independent verification of the fossil fuel emissions as its old organic carbon is radiocarbon-free (*Bozhinova et al.*, 2014; *Basu et al.*, 2016). Other chemical constituents like carbonyl sulfide (OCS) and solar induced chlorophyll fluorescence (SIF) could also be important additions in CTDAS. Inclusion of these tracers in the assimilation could enhance our understanding of carbon exchange, because variations in photosynthetic carbon uptake are recorded in atmospheric OCS and satellite SIF data (*Commane et al.*, 2015; *Yang et al.*, 2014).

Before we can interpret signals derived from these additional tracers, our aim for this paper is (1) to explain how the first multi-species CTDAS application works, with specific focus on the use of $\delta^{13}C$ and $CO_2$, henceforth the system named as CTDAS-C13 version 1, (2) to demonstrate its accuracy in solving the targeted optimization problem in comparison to observations, (3) to test the sensitivity of the system to the introduced nonlinearity arising from simultaneous optimization of terrestrial total $CO_2$ and $^{13}CO_2$ fluxes, and (4) to verify our new estimates of carbon and isotope exchange with independent drought index data.





## 2 Methodology

We present the atmospheric $\delta^{13}$C budget (Section 2.1) before proceeding to describe the integration of $\delta^{13}$C within our new multi-species data assimilation framework CTDAS-C13 (Section 2.2). We then briefly describe the prior estimates and the observational network used (Section 2.3). Finally, we give a brief description of the different inversion experiments (Section

2.4). The methodology presented here is based on Section 4.2 of the lead author's PhD dissertation (*Van der Velde*, 2015).

### 2.1 Atmospheric $\delta^{13}$C budget

The use of $\delta^{13}$C observations alongside $CO_2$ observations constitute a useful change to the traditional $CO_2$-only CTDAS application, as it provide an additional constraint on carbon surface fluxes and isotope exchange processes in plants. The rationale behind this is that the $^{13}CO_2$ and $^{12}CO_2$ contents in the atmosphere are affected through the same $CO_2$ pathways

from land and ocean surfaces. There are, however, specific processes that change the $^{13}CO_2$ exchange fluxes slightly differently from $^{12}CO_2$ fluxes. We can write a global mass balance for atmospheric $\delta^{13}$C ($\delta_a$) so that the different isotopic processes are explicitly defined and dependent on total $CO_2$ fluxes (see *Tans et al.*, 1993, for the derivation of Eq. 1). We can then identify the (1) emission forcing terms, (2) net exchange isotope forcing terms, and (3) gross-flux isodisequilibrium forcing terms:

$$
\begin{aligned}
C_a \tfrac{\mathrm{d}}{\mathrm{dt}} \delta_a \;=\; & F_{ff}\left(\delta_{ff} - \delta_a\right) & + \;\; & F_{fire}\left(\delta_{fire} - \delta_a\right) & & \text{[emission forcing terms]} \\
+ \;\; & N_b \epsilon_{ph} & + \;\; & N_o \epsilon_{ao} & & \text{[net exchange isotope terms]} \\
+ \;\; & F_{ba}\left(\delta_b - \delta_b^{eq}\right) & & & & \text{[terrestrial isodisequilibrium forcing terms]} \\
+ \;\; & F_{oa}\left(\delta_a^{eq} - \delta_a\right) & & & & \text{[ocean isodisequilibrium forcing term],}
\end{aligned}
\tag{1}
$$

where $C_a$ is the total carbon content [unit mol or mass] in the atmosphere (in the form of $CO_2$). The subscripts ba and oa denote the direction of the one-way gross fluxes [unit mol or mass per unit time]. For example, $F_{ba}$ refers to the respiratory release of $CO_2$ from terrestrial biosphere to atmosphere. The isotopic ratios of $^{13}$C/$^{12}$C are expressed as $\delta_{xx}$ [‰], where the subscripts refer to the signature in biosphere vegetation and soils (b), in biomass burning flux (fire), or in the fossil fuel emission flux (ff). The signature $\delta_a^{eq}$ depicts the isotopic ratio of $CO_2$ that is in equilibrium with the ocean surface and $\delta_b^{eq}$

depicts the ratio in the terrestrial biosphere that would be in isotopic equilibrium with the current atmosphere, which is more depleted in $^{13}CO_2$ than when the biomass was formed years ago. $N_b$ and $N_o$ refer to net exchange fluxes (gross release minus gross uptake) of $CO_2$, and $F_{ff}$ and $F_{fire}$ are the fossil fuel and biomass burning $CO_2$ emissions, respectively.

The terrestrial (photosynthetic) isotopic discrimination in Eq. 1 is expressed as $\epsilon_{ph} = (\delta_b^{eq} - \delta_a) \approx -\Delta_{ph}$ [‰], and can be derived from a $CO_2$ gradient-weighted average of different isotope fractionation effects during the transfer of $CO_2$ molecules

from the canopy air until their reaction with the enzyme Ribulose-1,5-bisphosphate (Rubisco) in the chloroplasts of the plant leaf. There are two main fractioning effects along this pathway; the plant fractionates with $\Delta_s = 4.4$‰ when $CO_2$ diffuses from leaf boundary through leaf stomata, and with $\Delta_f = 28$‰ during carboxylation. Smaller fractionation effects occur during diffusion between canopy air and leaf boundary ($\Delta_b = 2.9$‰), and during dissolution of $CO_2$ in mesophyll water ($\Delta_{diss} = 1.1$‰) and transport to chloroplasts ($\Delta_{aq} = 0.7$‰). The parameterization of $\Delta_{ph}$ for $C_3$ plants has been described by *Farquhar*





*et al.* (1982) takes the following form as in *Suits et al.* (2005):

$$\Delta_{\mathrm{ph}} = \Delta_{\mathrm{b}} \left( \frac{c_{\mathrm{a}} - c_{\mathrm{s}}}{c_{\mathrm{a}}} \right) + \Delta_{\mathrm{s}} \left( \frac{c_{\mathrm{s}} - c_{\mathrm{i}}}{c_{\mathrm{a}}} \right) + (\Delta_{\mathrm{diss}} + \Delta_{\mathrm{aq}}) \left( \frac{c_{\mathrm{i}} - c_{\mathrm{c}}}{c_{\mathrm{a}}} \right) + \Delta_{\mathrm{f}} \left( \frac{c_{\mathrm{c}}}{c_{\mathrm{a}}} \right), \tag{2}$$

where $c_{\mathrm{a,s,i,c}}$ represent $CO_2$ partial pressures in canopy air space, leaf boundary layer, stomatal cavity and in the chloroplasts, respectively. The overall discrimination $\Delta_{\mathrm{ph}}$ value reflects mostly the fractionation step with the highest resistivity (*O'leary*,

1988). For example, during a drought when the leaf's stomatal conductance is lowered in an attempt to prevent evaporative water loss, the diffusive $\Delta_{\mathrm{s}}$ is the most limiting factor, resulting in a lower overall $\Delta_{\mathrm{ph}}$. The opposite happens under more favorable environmental conditions when stomatal aperture is higher and carboxylation is the limiting factor, resulting in a higher overall $\Delta_{\mathrm{ph}}$.

The overall discrimination leaves the atmosphere relatively enriched and plants relatively depleted in $^{13}C$. $C_3$ plants are de-

pleted in $^{13}C$ by approximately $-20\,‰$ relative to the atmosphere and $C_4$ by approximately $-4\,‰$ as they can assimilate $^{13}CO_2$ more efficiently with Rubisco. $C_4$ photosynthesis is essentially a more complex form of carbon fixation than $C_3$ photosynthesis as it shields Rubisco in the bundle sheath cells from wastefully binding with oxygen rather than carbon dioxide.

In addition to discrimination effects during photosynthetic uptake, we also need to account for isotopic enrichment of the atmosphere through respiratory release of carbon with a heavier isotopic signature after spending from one year to several

decades or more in the plant and soil organic matter. This respiratory part will still enrich the atmosphere with $^{13}CO_2$ even if net $CO_2$ uptake equals zero (*Ciais et al.*, 1995), and we refer to it as the terrestrial isodisequilibrium flux in Eq. 1.

Discrimination associated with the dissolution of $CO_2$ in ocean water (*Zhang et al.*, 1995) is much smaller and spatially homogeneous ($\epsilon_{\mathrm{ao}} = -2\,‰$) than in the terrestrial biosphere. The difference between ocean and land discrimination provide an additional constraint on the net fluxes has already been demonstrated in previous studies (e.g., *Keeling et al.*, 1989; *Tans*

*et al.*, 1993; *Ciais et al.*, 1995; *Fung et al.*, 1997; *Rayner et al.*, 2008). We also have to account for isotopic disequilibrium that exists between the atmosphere and oceans. This isodisequilibrium flux is associated with the out-gassing of $CO_2$ from the ocean waters, and has globally an enriching tendency on the $\delta_{\mathrm{a}}$ signatures.

Besides the land and ocean discrimination and disequilibrium forcing terms we have two additional terms in Eq. 1. Firstly, there are $CO_2$ emissions due to combustion of fossil fuels, which have a distinct isotopic signature depending on the organic

fuel type, but globally its signature is approximately $\delta_{\mathrm{ff}} = -30\,‰$. Secondly, there are $CO_2$ emissions due to biomass burning, where $\delta_{\mathrm{fire}}$ bears the signature of the $^{13}CO_2$ and $^{12}CO_2$ fluxes of $F_{\mathrm{fire}}$, which is typically the signature of burnt leaf foliage, woody tissue and the aboveground litter (*Van der Velde et al.*, 2014).

## 2.2 CTDAS-C13

We followed the method presented by *Peters et al.* (2005) for designing the joint $CO_2$ and $\delta_{\mathrm{a}}$ data assimilation architecture.

Similar to the CarbonTracker Data Assimilation Shell (CTDAS) v1.0 discussed in detail by *Van der Laan-Luijkx et al., in prep.* (2017), we aim to close the $CO_2$ budget through fluxes from fossil fuel combustion, biomass burning, and net exchange fluxes from the terrestrial biosphere and oceans. In addition, we also intend to simultaneously close the $^{13}CO_2$ ($^{13}C_{\mathrm{a}}$) budget using the same set of $CO_2$ fluxes. Isotopic signatures themselves are not conserved quantities, therefore we calculate conserved mole



fractions of $CO_2$ and $^{13}CO_2$ in our transport model, which we can sample at designated locations and time to calculate $\delta_a$. The combined set of balance equations [unit $\mathrm{mol}$ per unit $\mathrm{time}$] takes the following form:

$$\frac{\mathrm{d}}{\mathrm{dt}}C_a \quad = \quad F_{ff} + F_{fire} + \lambda_b N_b + \lambda_o N_o, \tag{3}$$

$$\frac{\mathrm{d}}{\mathrm{dt}}{}^{13}C_a \quad = \quad {}^{13}F_{ff} + {}^{13}F_{fire} + {}^{13}N_b + {}^{13}N_o. \tag{4}$$

After some manipulation of Eq. 3 and 4, by following *Tans et al.* (1993), we obtain:

$$\frac{\mathrm{d}}{\mathrm{dt}}{}^{13}C_a \quad = \quad F_{ff}R_{ff} + F_{fire}R_{fire} + \lambda_b N_b \left(\lambda_{discr}\epsilon_{ph}/1000 + 1\right)R_a + \lambda_o N_o \left(\epsilon_{ao}/1000 + 1\right)R_a$$
$$+ \quad D_b + D_o. \tag{5}$$

The $^{13}C_a$ balance equation is now a close analog of Eq. 1, because $^{13}C_a$ is a function of discrimination, $N_b$ and $N_o$, and isodisequilibrium fluxes. The $R$ values depict the isotopic ratio of $^{13}CO_2/CO_2$ in the atmosphere ($R_a$), in fossil fuel ($R_{ff}$) and biomass burning emissions ($R_{fire}$), and their values are approximately 0.011. The isodisequilibrium fluxes from land and ocean surfaces are here simply shown as $D_b$ and $D_o$, respectively. The term $\left(\lambda_{discr}\epsilon_{ph}/1000 + 1\right)$ represents the optimized ratio between the isotopic signature in the photosynthetic flux and atmosphere ($R_{ph}/R_a$), and ranges between 0.980 and 0.996 depending on the prior $\epsilon_{ph}$ and discrimination scaler $\lambda_{discr}$. The term $\left(\epsilon_{ao}/1000 + 1\right)$ represents the ocean flux ratio and is held constant at 0.998 assuming $\epsilon_{ao} = -2\text{‰}$. The parameters $\lambda_b$ and $\lambda_o$ represent the linear scaling factors for each week and ecosystem region (ecoregion) to adjust the net carbon exchange over land and ocean surfaces, respectively. For land, the scaling factor is associated with one scalar per ecoregion based on the *Olson* (1985) land use classification following *Peters et al.* (2005, 2007) (Fig. 1). The terrestrial biosphere is further divided into 11 larger geographical areas also known as TransCom regions (*Gurney et al.*, 2002). Like in the early CT releases, each of the 11 TransCom land regions contains a maximum of 19 ecoregion types (Fig. 2) and the ocean is divided into 30 large basins encompassing large-scale ocean circulation features. This gives a maximum of 239 (=11·19+30) different scaling factors each week (*Peters et al.*, 2007). The new parameter is $\lambda_{discr}$, which is used to scale a maximum of 209 terrestrial discrimination parameters per week. They are associated with the same 1 $\times$ 1 degree ecoregions as the terrestrial fluxes. Note that the maximum number of scalable land parameters is in reality ~130, and not 209, because not each land region contains all 19 ecoregion types.

The terrestrial net exchange term in Eq. 5 ($\lambda_b N_b \left(\lambda_{discr}\epsilon_{ph}/1000 + 1\right)R_a$) includes two multiplicative scaling factors, making the required solution nonlinear. This poses a potential problem where variations in net exchange and discrimination are cancelling each other out to such a degree that it leads to low signal-to-noise, especially in discrimination. This is further investigated in Section 3.2. The fossil fuel combustion, biomass burning, and terrestrial and ocean isodisequilibrium fluxes all remain fixed a priori estimates. We describe in Section 3.1 the tuning of the latter disequilibrium fluxes to close the long-term mean balance of $\delta^{13}C$ in our system.

The scaling factors $\lambda_b$, $\lambda_o$, and $\lambda_{discr}$ are the unknowns that are combined in state vector **x** (with dimension s), for which we will try to find an optimal solution by minimizing a quadratic cost function. In this function there is a balance between



information drawn from the observation vector $\mathbf{y}$ (with dimension m) with a covariance $\mathbf{R}$ (m × m) and prior knowledge from the state vector $\mathbf{x}_\mathrm{p}$ (s) with a covariance $\mathbf{P}$ (s × s):

$$J = (\mathbf{y} - H(\mathbf{x}))^T \mathbf{R}^{-1} (\mathbf{y} - H(\mathbf{x})) + (\mathbf{x} - \mathbf{x}_\mathrm{p})^T \mathbf{P}^{-1} (\mathbf{x} - \mathbf{x}_\mathrm{p}). \tag{6}$$

The observation operator $H$ (m) represents the atmospheric transport model that propagates the surface fluxes from Eqs. 3 and 5 and samples accordingly the mole fractions of $CO_2$ and $^{13}CO_2$ at the same location and moment as the observations $\mathbf{y}$.

The solution for $\mathbf{x}$ that minimizes J is (*Tarantola*, 2005):

$$\mathbf{x} = \mathbf{x}_\mathrm{p} + \mathbf{K} \cdot [\mathbf{y} - H(\mathbf{x}_\mathrm{p})], \tag{7}$$

where $\mathbf{K}$ represents the Kalman gain matrix (*Peters et al.*, 2005). Eq. 7 can be expressed in terms of $\lambda$ (posterior scaling factor), $\lambda_p$ (prior scaling factor) and separate measurements of $CO_2$ ($c$) and $\delta^{13}C$ ($\delta$) with dimensions (j) and (k), respectively:

$$\begin{pmatrix} \lambda_{\mathrm{bio}1} \\ \cdot \\ \cdot \\ \lambda_{\mathrm{bio}209} \\ \lambda_{\mathrm{oce}210} \\ \cdot \\ \cdot \\ \lambda_{\mathrm{oce}239} \\ \lambda_{\mathrm{discr}240} \\ \cdot \\ \cdot \\ \lambda_{\mathrm{discr}448} \end{pmatrix} = \begin{pmatrix} \lambda_{P_{\mathrm{bio}1}} \\ \cdot \\ \cdot \\ \lambda_{P_{\mathrm{bio}209}} \\ \lambda_{P_{\mathrm{oce}210}} \\ \cdot \\ \cdot \\ \lambda_{P_{\mathrm{oce}239}} \\ \lambda_{P_{\mathrm{discr}240}} \\ \cdot \\ \cdot \\ \lambda_{P_{\mathrm{discr}448}} \end{pmatrix} + \mathbf{K} \cdot \left[ \begin{pmatrix} c_1 \\ c_2 \\ c_3 \\ c_4 \\ \cdot \\ c_j \\ \delta_1 \\ \delta_2 \\ \delta_3 \\ \delta_4 \\ \cdot \\ \delta_k \end{pmatrix} - H \begin{pmatrix} \lambda_{P_{\mathrm{bio}1}} \\ \cdot \\ \cdot \\ \lambda_{P_{\mathrm{bio}209}} \\ \lambda_{P_{\mathrm{oce}210}} \\ \cdot \\ \cdot \\ \lambda_{P_{\mathrm{oce}239}} \\ \lambda_{P_{\mathrm{discr}240}} \\ \cdot \\ \cdot \\ \lambda_{P_{\mathrm{discr}448}} \end{pmatrix} \right]. \tag{8}$$

In state vectors $\mathbf{x}$ and $\mathbf{x}_\mathrm{p}$ the scaling factors for terrestrial discrimination are appended after the flux scaling factors. Similarly, in the observation vectors $\mathbf{y}$ and $H(\mathbf{x}_\mathrm{p})$ the $\delta^{13}C$ observations are appended after the $CO_2$ observations. The $\mathbf{K}$ matrix determines how much a scaling factor needs to change given a set of $CO_2$ and $\delta^{13}C$ measurements. The matrices $\mathbf{R}$ and $\mathbf{P}$ modulate whether observations or bottom-up estimates are given more weight to the solution.

The $\mathbf{P}$ matrix contains 448 × 448 elements in total and is shown in Fig. 3. The first 209 × 209 element block contains the land flux uncertainties per ecoregion and their spatial correlations. The second 30 × 30 element block contains the ocean flux uncertainties per ocean basin. We gave the land scalars and the ocean scalars a maximum uncertainty of 80 % and 100 % along the diagonal, respectively as in earlier CarbonTracker releases. The third 209 × 209 element block contains the terrestrial discrimination scalars with a maximum uncertainty of 20 % along the diagonal with an identical spatial correlation structure as applied to the terrestrial flux uncertainty scalars. This implies that we can scale $\epsilon_\mathrm{ph}$ by a factor of $1.0 \pm 0.2$, and thus for a typical $C_3$ plant ($\epsilon_\mathrm{ph} = -20\text{‰}$) the mean and uncertainty lies around $-20 \pm 4\text{‰}$. Furthermore, there is covariation between ecoregions of nearby TransCom regions, e.g., between North America boreal and temperate regions, and between Europe and Eurasian regions. We did not allow covariances between net exchange and discrimination in order to give the parameters enough freedom in the solution.

The covariance structure of $\mathbf{R}$ is similar to $CO_2$-only CTDAS, but is extended with additional uncertainties in $\delta^{13}C$ observations. These expected uncertainties quantify our ability to simulate observations given the uncertainty in atmospheric transport modeling and measurement errors. Section 2.3.5 gives an overview of the used uncertainties for each observation category.




With this inversion framework in place CTDAS-C13 progresses in a similar manner as the traditional $CO_2$-only CTDAS. For each week the set of unknowns in the state vector are updated in a cycle that contains two steps. First there is a forecast step, which is driven by our fluxes and current background state vector $\mathbf{x}_p$ to forecast an ensemble of $CO_2$ and $^{13}CO_2$ mole fractions 5 weeks ahead in time. This is followed by an analysis step to determine the new state of the system with Eq. 8

such that it is consistent with the observations for the current week of the cycle. The analyzed state is propagated to the next cycle using the same model as *Peters et al.* (2007, Eq. 1 of Supp. Material), and with this new state a new cycle begins with another forecast step to forecast a new ensemble of the background state 5 weeks ahead in time, now with an additional set of observations from a new week. The ensemble for each tracer is created from 150 ensemble members to provide a Gaussian probability density function of the state vector.

The simulation of atmospheric transport is provided by the two-way nested global transport model TM5 release 3 (*Krol et al.*, 2005). This application simulates the atmospheric transport of $CO_2$ and $^{13}CO_2$ at global $6 \times 4$ degree resolution, with no nesting. It is driven by 3-hourly meteorological output from ECMWF ERA-interim reanalysis (*Dee et al.*, 2011). All the $CO_2$ and $^{13}CO_2$ flux fields provided to the model are in units of $\mathrm{mol\,CO_2\,m^{-2}\,s^{-1}}$ and $\mathrm{mol\,^{13}CO_2\,m^{-2}\,s^{-1}}$, respectively. Atmospheric concentrations of $CO_2$ and $^{13}CO_2$ are calculated as mole fractions in $\mathrm{mol\,mol^{-1}}$. Signatures of $\delta^{13}C$ are computed to

the relative per mil value using the following conversion formulation in order to facilitate comparison with observations:

$$\delta^{13}C = \left( \frac{R}{R_{\mathrm{ref}}} - 1 \right) \cdot 1000, \tag{9}$$

where $R_{\mathrm{ref}}$ is the VPDB reference ratio adopted for $^{13}CO_2/(^{12}CO_2 + {}^{13}CO_2)$, which is 0.011112 (*Tans et al.*, 1993). $R$ is the ratio of simulated mole fractions $^{13}CO_2/CO_2$.

## 2.3   Prior estimates and observations

### 2.3.1   Terrestrial biosphere fluxes

The terrestrial first-guess net $CO_2$ exchange ($N_b$) and fire ($F_{\mathrm{fire}}$) estimates were calculated in the Simple-Biosphere Carnegie-Ames Stanford Approach model (SiBCASA, *Schaefer et al.*, 2008) on a $1 \times 1$ degree grid on a $10\,\mathrm{min}$ time resolution and were further processed into 3-hourly mean fluxes to serve as input for CTDAS-C13. SiBCASA is a biogeochemical model that calculates carbon, isotope, water, and energy exchange fluxes. It is driven by data 3-hourly ECMWF ERA-interim meteorology,

designed with a semi prognostic leaf pool to track seasonal plant phenology, and it uses GFED4 daily burned area disturbances to calculate fire fluxes at a fine temporal resolution (*Van der Velde et al.*, 2013, 2014). The model incorporates $C_3$ and $C_4$ plant types with their own photosynthesis calculations and 12 different aggregated ecosystems according to *Olson* (1985). Respiratory $CO_2$ release from the plant and soil is calculated in the CASA part of the model using 13 biogeochemical pools with environment-influenced turnover rates (*Schaefer et al.*, 2008).



### 2.3.2 Ocean fluxes

The ocean first-guess net $CO_2$ exchange ($N_o$) estimates derive from ocean inversions from *Jacobson et al.* (2007). These long term estimates are combined with the quadratic gas-transfer velocity from 3-hourly ECMWF ERA-interim wind fields (*Wanninkhof*, 1992) to create fluxes on a $1 \times 1$ degree grid at a 3-hourly temporal resolution. An additional trend was applied

to the fluxes to ensure that increases in anthropogenic uptake are proportional to increases to atmospheric $CO_2$ levels. (see: http://www.esrl.noaa.gov/gmd/ccgg/carbontracker)

### 2.3.3 Fossil fuel emissions

Fossil fuel $CO_2$ emissions ($F_{ff}$) were made available on $1 \times 1$ degree grid at a monthly temporal resolution. They are derived from a combination of databases: EDGAR4.2, CDIAC, and BP statistics. (see: http://www.esrl.noaa.gov/gmd/ccgg/

carbontracker)

### 2.3.4 Isotope and disequilibrium fluxes

To calculate the fluxes of $^{13}CO_2$ from land surfaces we used the photosynthetic discrimination parameterization (Eq. 2) for $C_3$ plants in the SiBCASA model (*Van der Velde et al.*, 2014). The weighted leaf level value for $C_3$ discrimination is typically 19.0‰, and given the more efficient $CO_2$ bonding with the Rubisco enzyme $C_4$ discrimination is 4.4‰ (*Still et al.*, 2003; *Suits*

*et al.*, 2005). Given the dominance of $C_3$ plant growth (70 % of global GPP) the global mean discrimination in SiBCASA has been estimated at $\Delta_{ph} = 15.2$‰. SiBCASA's spatial heterogeneity of land discrimination is shown in Fig. 4. It reflects the land use distribution and the environmental forcing. Large discrimination values can be found in the temperate regions, the boreal forests, and in the humid environments such as the tropical rain forests in South America, Africa and South East Asia. Small discrimination values can be found in the United States corn belt and in the dry climate regions such as the African savannas

and Australian grasslands, where there is abundance of $C_4$ plant growth. More subtle variations in $\Delta_{ph}$ in $C_3$ dominant regions are driven by differences in environmental conditions (e.g., humidity, groundwater availability, and light intensity). Weekly 1 $\times$ 1 degree fields for $\Delta_{ph}$ were used to map the regular 3-hourly net $CO_2$ fluxes to $^{13}CO_2$ fluxes:

$$[\text{terrestrial net } ^{13}C \text{ exchange term}] = \lambda_b N_b \left( \lambda_{discr} \epsilon_{ph} / 1000 + 1 \right) R_a, \tag{10}$$

where $\epsilon_{ph}$ is derived from SiBCASA's $\Delta_{ph}$ output. Their relation is straightforward:

$$\epsilon_{ph} = -\Delta_{ph} \tag{11}$$

For the calculation of $^{13}CO_2$ biomass burning flux we assumed $R_{fire}$ to be very close to the signature of newly assimilated photosynthates, i.e.:

$$^{13}F_{fire} = F_{fire} \left( \lambda_{discr} \epsilon_{ph} / 1000 + 1 \right) R_a. \tag{12}$$





The $^{13}CO_2$ fossil fuel emissions are calculated with $R_{ff} = 0.0107786$, given that the global mean value of $\delta_{ff}$ is equal to $-30\,‰$:

$$^{13}F_{ff} = F_{ff}R_{ff}. \tag{13}$$

Note that we did not vary $\delta_{ff}$ for different fuel types in this version of CTDAS-C13, but such variability could be included in
the future based on the work of *Andres et al.* (2000).

The ocean discrimination parameter $\epsilon_{ao}$ is assumed to be constant at $-2\,‰$, as in many comparable studies (e.g., *Tans et al.*, 1993; *Ciais et al.*, 1995; *Alden et al.*, 2010). The regular 3-hourly net $CO_2$ fluxes were mapped to $^{13}CO_2$ fluxes:

$$[\text{ocean net } ^{13}\text{C exchange term}] = \lambda_o N_o \left(\epsilon_{ao}/1000 + 1\right) R_a, \tag{14}$$

The isodisequilibrium fluxes ($D_b$ and $D_o$, in mol $^{13}CO_2$ m$^{-2}$ s$^{-1}$) were made available on a monthly $1 \times 1$ degree reso-
lution. $D_b$ is calculated using SiBCASA's gross natural respiratory flux scaled with isotopic disequilibrium of the terrestrial
biosphere with the current atmosphere, i.e., $F_{ba} \left(\delta_b - \delta_b^{eq}\right)$. Because fossil fuel emissions add isotopically depleted $CO_2$ to the
atmosphere, the biosphere signature $\delta_b$ follows with a time lag dependent on the residence time of carbon in the vegetation
and soils. That implies $\delta_b$ is larger than $\delta_b^{eq}$, which is the biosphere signature that is in equilibrium with the current atmosphere
(*Tans et al.*, 1993). $D_b$ has a positive tendency on atmospheric $\delta^{13}C$ as carbon originating from different SiBCASA pools is
older and more enriched in $^{13}C$ than the isotopic signature of recently fixed photosynthates. The SiBCASA pool configuration
is described in detail in *Van der Velde et al.* (2014).

$D_o$ is calculated from the out-gassing flux of $CO_2$ scaled with the isotopic disequilibrium of the ocean surface with the
current atmosphere, i.e., $F_{oa} \left(\delta_a^{eq} - \delta_a\right)$. The $\delta_a^{eq}$ term is determined from a global network of $\delta^{13}C$ measurements in dissolved
inorganic carbon (*Gruber et al.*, 1999). $F_{oa}$ is parameterized as a function of surface ocean partial pressure of $CO_2$ and wind-
speed after *Takahashi et al.* (2009). Windspeed and solubility are assumed to remain constant year-to-year. The disequilibrium
fluxes are positive from the equator to approximately 60 degrees of latitude in both directions and are negative beyond that.

### 2.3.5 Observations

Observations of $CO_2$ from a wide range of research laboratories are bundled in Observation Package (ObsPack) version 1.0.3
and observations of $\delta^{13}C$ from the INSTAAR Stable Isotope Lab are bundled in version 1.0.0. These are data products that
include the provider's original data and metadata reformatted into the ObsPack framework (*Masarie et al.*, 2014).

From the available $CO_2$ observations, approximately 24,000 weekly flask measurements were used in the assimilation from
a fixed network of 58 surface sites. Another large set of 174,000 measurements came from 23 semi-continuous in-situ sites.
Most $CO_2$ measurements are obtained with a nominal precision of $\pm 0.1$ ppm. The remainder of sites and measurements (in-
cluding from aircraft or shipboard) were not used because of double records, and some measurements were kept for independent
checks. A small fraction was omitted as our model could not resolve certain locations at a coarse resolution.

For the multi-species inversions we also used 22,000 flask measurements of $\delta^{13}C$ from 53 different surface sites. A further
5,600 measurements from five different sites were obtained using programmable flask packages (PFP), which measure $\delta^{13}C$ at
a daily resolution. The isotope ratios are measured by dual inlet mass-spectrometry with a precision of $\pm 0.01\,‰$.



We determined observation uncertainties (model-data mismatch, or MDM) for each of the $\delta^{13}$C measurement sites in a heuristic manner based on earlier test inversions. These values are added to the diagonal of **R**. A too small error would give an unrealistic amount of confidence how well the model is expected to represent the measurement location during sampling but a too large error we would give very little confidence to the measurement representation.

The $\delta^{13}$C measurement sites were divided into different categories each with their own MDM value. As with $CO_2$ these categories were: land, mixed conditions, marine boundary layer (MBL), deep Southern hemisphere, and a special category for problem sites where forecast performance is poor. For each site we determined the innovation statistic $\chi^2$, which is a measure for how apt our applied uncertainty level is given the model-data fit. A $\chi^2$ value of 1.0 indicates that the simulated and expected total uncertainty are equal, lower values indicate overestimation of the uncertainty, and higher values underestimation. Table 1

gives a summary of the site categories used, together with the assigned MDM for $\delta^{13}$C and the category-average innovation $\chi^2$ determined from an inversion experiment. For the majority of sites the innovation values are between 0.7 and 1.3, i.e., around the ideal value of 1.0. For the $CO_2$ measurement sites we used a similar set of MDM values as in previous CarbonTracker releases.

## 2.4 Experiments

We performed four inversion experiments as summarized in Table 2. The simulation period covered the years 2000 through 2011, but our analyses focused on the period 2001-2011, i.e., we omitted the spinup year. As a benchmark we performed a traditional inversion to estimate the net carbon exchange fluxes of the ocean and land using only $CO_2$ observations, which we call TRAD−CO2. For the second inversion we added $\delta^{13}$C observations alongside $CO_2$ to constrain only the exchange fluxes, therefore we call this experiment TRAD−CO2C13. The experiment in which we estimated discrimination and fluxes

simultaneously is called NEW−CO2C13. This inversion is nonlinear because the discrimination scaling parameter is in the same multiplication term as the net flux scaling parameter. The fourth experiment was a linear inversion experiment where we estimated only the land discrimination parameter using $\delta^{13}$C data. We call this experiment NEW−2STEP because discrimination was solved in a second step after optimization of the net exchange fluxes. That means that ocean and land fluxes were derived from the optimized state vector and its covariance from the TRAD−CO2 inversion.

## 3 Results

## 3.1 Comparison to observations of $CO_2$ and $\delta^{13}$C from the global network

We first evaluate the global $CO_2$ and $\delta^{13}$C budgets simulated by our combination of fluxes as described in Section 2, to assess where we expect the largest changes in the optimization. As shown in Fig. 5a, the prior net exchange flux estimates and unscaled disequilibrium fluxes were not large enough to close the gap with the observed tracers, $CO_2$ and $\delta^{13}$C. The sum of the

flux arrows overestimated the annual $CO_2$ growth rate along the $x$-axis and overestimated $\delta^{13}$C depletion along the $y$-axis. In a traditional TRAD−CO2 inversion the estimated ocean and land fluxes closed the $CO_2$ budget along the $x$-axis. The leverage





in the net exchange fluxes was however not large enough to close the $\delta^{13}C$ budget along the $y$-axis as well. In an inversion that includes $\delta^{13}C$ observations, the gap in $\delta^{13}C$ would adjust the $CO_2$ flux magnitudes and ocean/land partitioning to unrealistic magnitudes in an effort to overcome the large offset between the simulated and observed $\delta^{13}C$ growth rate. Instead we chose to use scaled disequilibrium fluxes in our inversions in order to estimate land and ocean $CO_2$ flux magnitudes that remain close

to the results of other traditional carbon cycle budgeting studies (*Alden et al.*, 2010; *Van der Velde et al.*, 2013). We chose the disequilibrium fluxes to adjust because (1) the exact magnitudes of these terms are still unknown due to uncertainties in the carbon pool turnover, gross carbon fluxes and isotopic discrimination, and (2) these terms do not affect the $CO_2$ mass balance. It assured a closed mean $\delta^{13}C$ budget of our inversions without creating unrealistic carbon sinks over land and oceans (Fig. 5b). Most importantly, closing the climatological (11-year) budget allowed us to focus our study on interannual changes in the

net fluxes and photosynthetic discrimination.

We obtained the best fit with $\delta^{13}C$ data when the land and ocean disequilibrium flux were scaled by a factor of 1.2 without changing either their spatial patterns or time trends. This is consistent with recent double deconvolution studies where the global $\delta^{13}C$ balance was closed with a factor of 1.3 in land and ocean disequilibrium (*Alden et al.*, 2010). Our value was determined after assessing an ensemble of different sets of scaling numbers (ranging from 1.1 to 1.5) in a forward TM5

simulation, which was driven by the optimized net land and ocean flux estimates from the TRAD−CO2 experiment. This assured a closed multi-year $\delta^{13}C$ budget together with a closed multi-year $CO_2$ budget. As selection criteria we used (1) the 11-year mean Root-Mean-Square-Difference (RMSD) of a large selection of $\delta^{13}C$ sites and (2) the average bias between simulated and observed values. In the non-scaled disequilibrium simulation we obtained a RMSD of $0.165‰$ and a bias of $-0.110‰$ averaged over all sites. The optimal result was obtained with a scaling factor of 1.2, which reduced the RMSD to

$0.079‰$ and the mean bias to $-0.010‰$. Note that these scaling factors cannot be applied to other inversion studies because the disequilibrium scaling factors are tuned for this particular system and time period.

To demonstrate our procedure in terms of individual data sets, we refer to Fig. 6. After scaling the disequilibrium fluxes and using optimized net carbon exchange from the TRAD−CO2 inversion, time series of $\delta^{13}C$ at 32 of the 46 Northern hemisphere sites showed no remaining significant trend (sites where $p$-value $> 0.05$) in the summer residuals, and the residuals from the

trend lines were within or close to the MDM specified for our multi-species inversions. Some of the sites with remaining trends are located at great distances from large continental carbon sources and sinks, and exert little influence on the posterior $\lambda_{discr}$ parameter (e.g., CHR, GMI). Some of the other sites were assigned a large MDM (e.g., BAL, NWR, TAP) giving them less weight in the estimation of the posterior $\lambda_{discr}$ parameter. The collection of sites with remaining trends do not seem to have a systematic geographic pattern and are likely reflecting a change in local oceanic or biospheric isotope exchange, such as

must be the case for the Bermuda West (BMW, non-significant positive trend) and Bermuda East (BME, significant downward trend) site.

With the long-term trend of $\delta^{13}C$ appropriately captured, we proceeded to optimize NEE and $\Delta_{ph}$ with our new framework (NEW−CO2C13). We show that this inversion further reduced $\delta^{13}C$ residuals (Fig. 7a), without compromising (nor strongly improving) the fit to $CO_2$ (Fig. 7b) that we attained from the TRAD−CO2 inversion. In Fig. 7a the ratio of $\delta^{13}C$ RMSD



of NEW−CO2C13 to $\delta^{13}$C RMSD of the TRAD−CO2 inversions was at most sites smaller or equal to 0.95 (indicating a significantly higher accuracy of NEW−CO2C13 in form of bias and noise reduction):

$$\frac{\delta^{13}\text{C RMSD (NEW−CO2C13)}}{\delta^{13}\text{C RMSD (TRAD−CO2)}} < 0.95$$

In Fig. 7b, the ratio of $CO_2$ RMSD of NEW−CO2C13 to $CO_2$ RMSD of TRAD−CO2 was at most locations between
0.95 and 1.05. This suggests that the two atmospheric constraints applied are complementary, and there is no indication that the TRAD−CO2 results from CarbonTracker were inconsistent with $\delta^{13}$C measurements. This is an important prerequisite for a credible estimate of discrimination in our system. Furthermore, Fig. 7a shows a notable latitudinal divide in the reduction of $\delta^{13}$C RMSD, indicating the utility of NEW−CO2C13 in the Northern hemisphere due to the large availability of measurements and scalable discrimination parameters.

At sites like Alert (Nunavut, Canada) the NEW−CO2C13 inversion provided a better fit to the measured data than the TRAD−CO2 inversion (Fig. 8). The 11-year averaged $\delta^{13}$C residuals were close to zero for both inversions, as the disequilibrium flux was tuned specifically to prevent large residuals in a-priori simulated $\delta^{13}$C as described in Section 3.1. The $1\sigma$ standard deviation of the $\delta^{13}$C residuals at Alert were smaller in the NEW−CO2C13 inversion in comparison to TRAD−CO2, due to the additional optimization of $\Delta_{\text{ph}}$ alongside net exchange fluxes. The $CO_2$ residuals for Alert in Fig. 8 were for both
inversions almost identical.

### 3.2 Linear and nonlinear estimates of net carbon uptake and land discrimination

Simultaneously optimizing both $\lambda_{\text{discr}}$ and $\lambda_{\text{bio}}$ is inherently nonlinear and thus possibly problematic for our assimilation system, therefore we tested the validity of our approach in the NEW−CO2C13 inversion. We hypothesized that a region's net carbon uptake and discrimination would change in a similar fashion in the nonlinear inversion, as it would for a linear
inversion. The linear inversion experiment consisted of two consecutive steps: (1) the optimization of the net exchange fluxes using only $CO_2$ observations (TRAD−CO2) followed by (2) the estimation of the land discrimination parameter using only $\delta^{13}$C observations (NEW−2STEP). In the nonlinear NEW−CO2C13 inversion the optimization of fluxes and discrimination was done simultaneously. For net carbon uptake by vegetation we refer to Net Ecosystem Exchange, or NEE, defined as positive when $CO_2$ is taken up from the atmosphere. For plant isotope discrimination we refer to $\Delta_{\text{ph}}$ in per mil, which is defined as
positive.

As shown in Fig. 9 the 11-year mean NEE for the 11 land TransCom regions are very similar in the nonlinear NEW−CO2C13 and linear TRAD−CO2 inversions. Deviations are in the order of tens of teragrams, and within $1\sigma$ standard deviation of the flux interannual variability (IAV). Fig. 9 also shows the impact of $C_4$ photosynthesis on the mean TransCom aggregated $\Delta_{\text{ph}}$ values. In the boreal regions, where there is very little $C_4$ plant growth, the discrimination is at its maximum (approximately
20‰, 5‰ above the global average), but in regions where there is $C_4$ plant growth (e.g., due to agriculture in the United States or savannas in Africa) the mean $\Delta_{\text{ph}}$ values are lower (approximately 12-15‰). These regional patterns of $\Delta_{\text{ph}}$ imposed by SiBCASA (see also Fig. 4) are maintained by the NEW−CO2C13 inversion framework. Because we aimed to retrieve robust





temporal patterns of IAV, the most relevant indicators for the robustness of our nonlinear inversion approach are given by the correlation coefficients ($r$) between the two types of inversions. We calculated $r$ for NEE and $\Delta_{\mathrm{ph}}$ between the linear and nonlinear inversions. As the seasonal cycles in uptake and discrimination are largely dictated by the prior estimates, we removed them using a 3-month boxcar mean smooth curve fitting to obtain the anomalies relative to the seasonal trend. The NEE in

NEW−CO2C13 is very similar to the NEE in TRAD−CO2, as indicated by the high $r$-values (>0.96 for N=52·11 weeks) for all TransCom regions. The $r$ values are lower for $\Delta_{\mathrm{ph}}$, but still exceed 0.75 in the Northern hemisphere. The correlation is particularly high over North America Boreal, North America Temperate, and European regions. Smaller correlations are obtained in Tropical and Temperate South America and Tropical Asia. This is expected, however, as these regions typically suffer from a lack of observational constraints.

The linear NEW−2STEP inversion estimated the same large increase in discrimination IAV as in the nonlinear NEW−CO2C13 inversion for the Northern hemisphere in comparison to the first-guess estimate of SiBCASA (8-fold increase in standard deviation, see Table 3). In addition, we also found in both inversions a strong positive correlation between $\Delta_{\mathrm{ph}}$ and NEE on annual time scales ($r = 0.79$, $N$=9, $p$=0.001). In years when annual mean NEE is low (less carbon uptake) the $\Delta_{\mathrm{ph}}$ is low too (less discrimination), implying that stomata have partially closed, and vice versa. This correlation did not emerge in the TRAD−CO2

estimate based on atmospheric $CO_2$ observations alone, and it also did not emerge if $\delta^{13}C$ observations were additionally used in the TRAD−CO2C13 estimate, to estimate NEE but not $\Delta_{\mathrm{ph}}$. The SiBCASA terrestrial biosphere model that provides the first-guess NEE and $\Delta_{\mathrm{ph}}$ of our data assimilation framework based on commonly used drought response parameterizations, simulated neither the large IAV in NEE and $\Delta_{\mathrm{ph}}$ nor their strong correlation. It is evident from the NEW−2STEP inversion that changes in $\Delta_{\mathrm{ph}}$ and the correlation with NEE were driven by $\delta^{13}C$ observations, and were not a symptom of the systems

inability to separately estimate NEE and $\Delta_{\mathrm{ph}}$ variations. This suggests that the estimated IAV of $\Delta_{\mathrm{ph}}$ in the nonlinear inversion is truly a signal retrieved from $\delta^{13}C$ that would otherwise be aliased erroneously into the carbon fluxes or not retrieved at all.

### 3.3 Independent verification with drought indices

A closer inspection reveals that the reported correlation between the Northern hemisphere's NEE and $\Delta_{\mathrm{ph}}$ in Table 3 could indicate a moisture driven response at ecosystem level. We identified several moments of severe to extreme drought as char-

acterized by a Standardized Precipitation and Evaporation Index (SPEI, *Vicente-Serrano et al.*, 2010) below -1.0 that covered an extensive area of more than a $\mathrm{million\,km^2}$ in United States. These droughts are described in literature as the droughts (or heat waves) of summer 2002 (*Seager*, 2010; *Schwalm et al.*, 2012) and 2011 (*Long et al.*, 2013). The annual averaged maps of SPEI for 2001-2011 are shown in the top panel of Fig. 10 calculated for the Northern American Temperate TransCom domain. Independent of the SPEI drought index, we estimated changes in $\Delta_{\mathrm{ph}}$ and NEE over the same American domain

with the NEW−CO2C13 inversion using atmospheric $CO_2$ and $\delta^{13}C$ data (Fig. 10, middle and lower panels). A correlation between $\Delta_{\mathrm{ph}}$ and SPEI could only be established by applying an area weighting function to the SPEI index to give years that





experienced large and severe droughts the strongest association with reductions in $\Delta_{\mathrm{ph}}$. We used the following function for the Weighted Drought Index (WDI):

$$\mathrm{WDI} = \frac{\sum_{i=1}\left(\mathrm{SPEI[i]} \cdot \ \mathrm{Gridcell\text{-}area[i]}\right)}{\mathrm{Total\text{-}area}}.$$

In words, we sum over the product of the SPEI index and the grid cell surface area where SPEI is below -1.0 and subsequently we divide it by the total area of the TransCom domain. Hence, the WDI is an expression the drought in terms of the surface area that is affected. A larger drought surface area will result in a more negative WDI. Using this function we see that the lower values for $\Delta_{\mathrm{ph}}$ correspond strongly with years of low SPEI over large serried areas, indicating a temporal correlation between the SPEI variable and $\Delta_{\mathrm{ph}}$ (see correlation in Fig. 11: 95 % confidence interval of a two-sided distribution with 9 degrees of freedom, p=0.008, r=+0.75). The two largest anomalies ($> 1\sigma$ of 11-year IAV) in annual mean $\Delta_{\mathrm{ph}}$ correspond with low SPEI in 2002 and in 2011. A third notable drought as recorded in SPEI happened in 2006, and although carbon uptake was reduced, it did not amount to a significant signal in $\Delta_{\mathrm{ph}}$. Similar correlations do exist over other parts of the Northern hemisphere in our inversion solution. For instance, severe droughts in Western Europe (2003) and Russia (2010) lowered the discrimination by 1.0 ‰, and exceeded more than $1\sigma$ standard deviation of its 11-year IAV (not shown).

In addition, in years when $\Delta_{\mathrm{ph}}$ is low, the annual mean NEE tends to be low too, possibly as a result of reduced GPP. This implies that leaf stomata have partially closed and therefore affecting both $\Delta_{\mathrm{ph}}$ and carbon uptake from photosynthesis. The reduction of the optimized net carbon sink for North America is 100-400 $\mathrm{Tg\,C\,yr}^{-1}$ during the drought years of 2002, 2006 and 2011 (in comparison to their surrounding years).

These correlations that are averaged over continent sized areas do however breakdown on smaller scales. At regional scales we observed a partial misallocation of the model adjustments of NEE and $\Delta_{\mathrm{ph}}$ in comparison to SPEI. This is largely a consequence of our limited capacity to monitor $CO_2$ and $\delta^{13}C$. For example, for North America Temperate 2002, where the drought index was negative over the mountain states, the impact on the carbon cycle was strongest over the eastern forests of the United States. In these forest ecosystems $CO_2$ exchange is much stronger than over the mountains, and hence their impact on atmospheric $\delta^{13}C$ as well.

Notice that the prior net carbon sink is underestimated in comparison to the optimization because SiBCASA assumes a near steady state between between GPP and TER (Fig. 10). SiBCASA was in fact able to simulate small carbon uptake anomalies during the reported droughts using its own environmental response parameterizations. However, it lacked substantial amount of interannual variability in NEE and $\Delta_{\mathrm{ph}}$ nor a strong correlation of $\Delta_{\mathrm{ph}}$ with SPEI (Fig. 11). This suggests a potential absence of an important coupling between the hydrology and carbon discrimination processes in the model.

## 4 Discussion and conclusions

We developed a new application of the CarbonTracker data assimilation system that simulates two atmospheric tracers simultaneously: $CO_2$ and the $\delta^{13}C$ isotope signature of $CO_2$. We used measurements of both tracers to optimize the net ocean and



land carbon exchange fluxes and the land discrimination parameter $\Delta_{\mathrm{ph}}$. The annual reductions in $\Delta_{\mathrm{ph}}$ were up 0.75‰ and exceeded the $1\sigma$ standard deviation of the IAV over 11 years in the North American domain (16.4±0.3‰). We interpret these negative anomalies in $\Delta_{\mathrm{ph}}$ as possible reductions of the intercellular $CO_2$ levels and relative increases of the intercellular $^{13}CO_2/^{12}CO_2$ ratio, resulting from stomatal closure due to drought stress at the leaf level. This is the most plausible explana-

tion as most other factors that affect $\Delta_{\mathrm{ph}}$ either (a) are included a-priori in SiBCASA biosphere model, such as the effects of IAV in strength of photosynthesis over $C_3$ and $C_4$ vegetation, or the variations in mesophyll conductance are (b) not expected to vary much from year-to-year, such as ecosystem composition, or (c) would enhance the intercellular $CO_2$ levels (and thus $\Delta_{\mathrm{ph}}$) rather than reduce it, such as increased radiance of the leaves under reduced cloud cover. This suggest the possibility that the impact of environmental stress on stomatal conductance and carbon uptake is much larger than currently simulated

by the widely used drought parameterizations in terrestrial biosphere models. These parameterizations are often derived from laboratory observations or plot-scale observations that often aggregate poorly over much larger scales. Our first results suggest that a data assimilation system that uses the global atmospheric $\delta^{13}C$ record, in concert with the $CO_2$ record, can offer new insights on large-scale drought dynamics of the coupled vegetation-atmosphere system.

It is unlikely our terrestrial biosphere model will reproduce the new large-scale atmospheric constraints on NEE and $\Delta_{\mathrm{ph}}$

with a simple adjustment of the currently used drought response parameterizations (such as stomatal conductance and soil water stress inhibition functions). We experimented with a different stomatal conductance model based on vapor pressure deficit (VPD, *Leuning*, 1995) rather than relative humidity as it was shown to better predict changes of the isotopic composition in tree rings (*Ballantyne et al.*, 2010). This modification however did not change the annual covariation between NEE and $\Delta_{\mathrm{ph}}$ in SiBCASA. In addition, modifications in the soil water stress function of SiBCASA, which impacts $\Delta_{\mathrm{ph}}$ through mesophyll

conductance (*Seibt et al.*, 2008) also had little impact on annual variations in $\Delta_{\mathrm{ph}}$. Instead, SiBCASA shows minimal dynamic range in the hydrological drivers of drought stress. This agrees with a separate assessment of the SiBCASA model with satellite observed soil moisture over Boreal EurAsia (*Van der Molen et al.*, 2016), and could have a number of causes. This includes (a) a too homogenous spatial domain with only one biome type, one soil type, and one soil moisture reservoir per $1 \times 1$ degree grid cell, (b) its hydrological scheme with relatively simple run-off and interception formulations, (c) a lack of realism in simulating

the latency of ecosystem recovery after a severe drought, or (d) even a possible misrepresentation of the effects of root-zone soil moisture stress, as was also diagnosed for the Amazon by *Harper et al.* (2010) for the closely related SiB model. There is also evidence that the conventional use of land cover types in biosphere models does not adequately describe the spatial variations of carbon exchange (*Bloom et al.*, 2016).

As with any data assimilation system, the number of available observations largely determines the assimilation system's

ability to retrieve meaningful signals. Our current method relies on atmospheric $\delta^{13}C$ anomalies that affect multiple monitoring sites at the same time due to low signal-to-noise at each site, but the network coverage over many parts of the world is still sparse. The increase of number of measurement sites, and the addition of $\delta^{13}C$ to many existing ones, particularly in sparsely populated areas could benefit CTDAS-C13 greatly. New measurement efforts are currently underway to improve our observational coverage in these sparsely sampled areas. Regular measurements of $CO_2$ from aircraft vertical profiles have

recently commenced at four different sites above the Amazon. These data have provided new insights on the carbon cycle





under drought conditions (*Gatti et al.*, 2014). These new measurements were successfully used in an application of CTDAS (*Van der Laan-Luijkx et al.*, 2015) and confirmed that the Amazonian $CO_2$ uptake by vegetation was indeed reduced during the severe 2010 drought. Furthermore, some coauthors are currently involved in a new collaborative effort to provide the first high-precision measurements of $\delta^{13}C$ and other isotopes in $CO_2$ from a large number of air samples collected over the Amazon

basin. Using an assimilation system similar to that described here, these data would bolster our ability to quantify seasonal to interannual changes in the Amazonian carbon balance and better understand the influence of drought stress on NEE.

The retrieved correlation between NEE and $\Delta_{\mathrm{ph}}$ in the Northern hemisphere was derived from atmospheric $\delta^{13}C$ observations through our new multi-species approach, and thereby provided new insights on the land-atmosphere coupling of water and carbon on continental and hemispheric scales. The unconstrained SiBCASA model does not show a large enough response

to drought both in terms of NEE and $\Delta_{\mathrm{ph}}$. The correlation between droughts and $\Delta_{\mathrm{ph}}$ over the North American Temperate domain (Fig. 10) can only be demonstrated after optimizing NEE and $\Delta_{\mathrm{ph}}$ by applying atmospheric $\delta^{13}C$ and $CO_2$ constraints together. We emphasize that the reported correlations remain robust and significant even when changing the atmospheric transport characteristics (i.e., convection fields from ECMWF ERA-Interim meteorology vs. default TM5 convection scheme), the optimization method (nonlinear vs. linear 2-step), and when changing the assumed model-data errors of our data assimilation

system.

A potential problem with the current framework is that we cannot account for changes in the terrestrial isodisequilibrium flux. In Eq. 1, we forced all missing isotopic variability into term $N_{\mathrm{b}}\epsilon_{\mathrm{ph}}$ without considering additional variability from the isodisequilibrium term. Photosynthetic discrimination is also responsible for a portion of the variability in the terrestrial isodisequilibrium flux (*Van der Velde et al.*, 2013), but the extent is hard to quantify. The $\delta_{\mathrm{b}}^{\mathrm{eq}}$ signature (i.e., the biosphere

signature that is in equilibrium with the atmosphere) is a function of the current $\delta_{\mathrm{a}}$ and $\Delta_{\mathrm{ph}}$, two quantities that ultimately exert influence on $\delta_{\mathrm{b}}$ as the isotopic signal carries through the series of carbon reservoirs (i.e., leaves, stems, roots, and ultimately the soils). The absence of direct adjustments to the disequilibrium flux could mean we aliased erroneously isotopic signals only onto the net flux term of the budget. In light of recent observational evidence, the variability of disequilibrium term might be of more importance than recently thought. *Bowling et al.* (2014) showed with $\delta_{\mathrm{a}}$ measurements that the disequilibrium

flux can become negative locally due to humidity induced changes in $\Delta_{\mathrm{ph}}$. Using a more simplified but physically consistent set of equations only based on gross fluxes (GPP and TER) to express the rate of change of $\delta_{\mathrm{a}}$ would eliminate the need for a disequilibrium term. This would on the other hand complicate the closing of the $CO_2$ budget as it necessitates a way to effectively separate these two gross fluxes.

It is worth mentioning that the carbon residence time in land ecosystems is highly uncertain, and therefore the gross $CO_2$

exchange as well. *Welp et al.* (2011) suggested that the current popular estimate of global GPP of $120\,\mathrm{Pg\,C\,yr^{-1}}$, which is also predicted by SiBCASA, may be a lower limit and could in reality be as large as $175\,\mathrm{Pg\,C\,yr^{-1}}$ to reflect faster turnover of carbon in the vegetation and soils. Such uncertainties were also underlined by *Carvalhais et al.* (2014) who found that higher precipitation rates are associated with faster carbon turnover, but that global modeled turnover is in fact often underestimated. We make a cautious conjecture that if GPP is in fact as large as claimed by *Welp et al.* (2011), and heterotrophic respiration is

large too, it will partly explain the current underestimation in the modeled disequilibrium fluxes, which are a function of TER



and ocean $CO_2$ outgassing. In this study we closed the gap with a predetermined scaling factor of 1.2 on the disequilibrium fluxes for oceans and land without assuming actual changes in GPP, TER or $\Delta_{ph}$. We could therefore benefit from a more integrated assimilation system where we are using atmospheric data to simultaneously optimize for terrestrial model parameters that exert influence on GPP, TER, carbon turnover. The CTDAS modular design (*Van der Laan-Luijkx et al., in prep.*, 2017)

5   makes it now more straightforward to develop and implement such additional improvements.

To conclude, this study showed there is significant potential to use atmospheric $CO_2$ and $\delta^{13}C$ data as constraints on plant NEE and isotopic discrimination using a multi-species assimilation platform. Signals that would otherwise be lost in a single tracer data assimilation system, such as the possibility of a drought driven covariation between isotope discrimination and NEE or the separation of GPP from NEE, can potentially be detected in the described multi-species application of CTDAS. Con-

10   tinued and additional measurements of atmospheric $\delta^{13}C$ and $CO_2$, especially in future assimilation systems where biosphere model parameters are directly optimized, should help us better understand the hydrological and biogeochemical interactions between the atmosphere and vegetation.

### Code availability

The CTDAS-C13 and TM5 source code are made available online as supplementary material on the GMD website. More

15   detailed model descriptions and information to run the code are available on the following websites: www.carbontracker.eu and tm.knmi.nl/index.php/Main_Page.

*Author contributions.* I.vdV, W.P and J.B.M designed the study. I.vdV, K.S, W.P. and M.vdM built the inverse and forward modeling frameworks. P.P.T, B.V., and J.W.C.W were responsible for the $\delta^{13}C$ and $CO_2$ measurement program. I.vdV performed the analysis and wrote the main text. All authors gave input on the final manuscript.

*Acknowledgements.* This study was financially supported by the Netherlands Organization for Scientific Research (NWO-VIDI: 864.08.012) and by the National Computing Facilities Foundation (NCF project SH-060) for the use of supercomputing facilities.



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

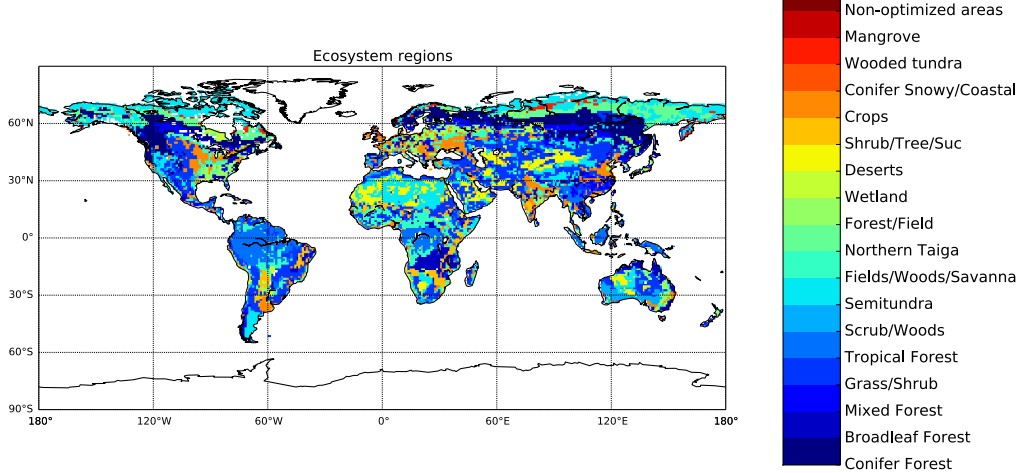

**Figure 1.** Global distribution of Olson ecosystem types.

**Table 1.** Summary of assigned $\delta^{13}$C model-data mismatch (MDM), the category-averaged and $1\sigma$ standard deviation of the innovation $\chi^2$, and number of sites per category.

| Site category | MDM (‰) | $\chi^2$ | # sites |
|---|---|---|---|
| land | 0.13 | 0.97±0.52 | 10 |
| mixed | 0.080 | 0.80±0.34 | 11 |
| marine boundary layer | 0.03 | 1.29±0.70 | 15 |
| deep Southern hemisphere | 0.03 | 1.22±0.44 | 7 |
| problem | 0.4 | 0.63±0.48 | 10 |





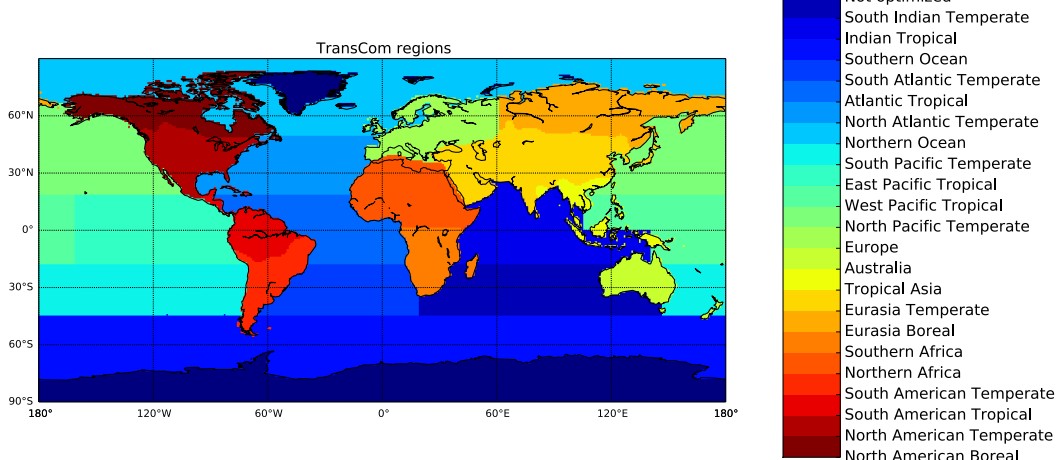

**Figure 2.** Earth's partitioning into 11 land regions and 11 ocean regions according to the TransCom project. The ocean regions are divided into 30 smaller basins (not shown) and the land regions can contain up to 19 different ecoregions as shown in Fig. 1.

**Table 2.** Summary of the four inversion experiments, the observations used, the optimized items (ocean and land fluxes, and $\Delta_{\mathrm{ph}}$), and their linearity. The prefix TRAD− refers to traditional, i.e., experiments that have been performed in the past in any way, shape or form. The prefix NEW− refers to a new type of inversions used in this publication. NEW−CO2C13 used the default multi-species CTDAS model setup as described in the Methodology, while NEW−2STEP solved for $\Delta_{\mathrm{ph}}$ using only $\delta^{13}$C data.

| Experiment | Observations | Optimization | Linear? |
|---|---|---|---|
| TRAD−CO2 | $CO_2$ | flux only | yes |
| TRAD−CO2C13 | $CO_2$ and $\delta^{13}$C | flux only | yes |
| NEW−CO2C13 | $CO_2$ and $\delta^{13}$C | flux and $\Delta_{\mathrm{ph}}$ | no |
| NEW−2STEP | $\delta^{13}$C | $\Delta_{\mathrm{ph}}$ only | yes |

**Table 3.** Northern hemisphere land net carbon uptake (NEE, [$\mathrm{Pg\,C\,yr^{-1}}$]) and land discrimination ($\Delta_{\mathrm{ph}}$, [‰]) 11-year mean estimates, and IAV ($\pm 1\sigma$ standard deviation) from SiBCASA (prior) and the four inversion experiments. The last line gives the correlation coefficient $r$ between 11 annual mean NEE and $\Delta_{\mathrm{ph}}$ values.

| | Prior | TRAD−CO2 | TRAD−CO2C13 | NEW−CO2C13 | NEW−2STEP |
|---|---|---|---|---|---|
| NEE | 0.22±0.28 | 2.44±0.46 | 2.65±0.49 | 2.58±0.46 | 2.44±0.46 |
| $\Delta_{\mathrm{ph}}$ | 18.1±0.02 | 18.1±0.02 | 18.1±0.02 | 18.2±0.17 | 18.3±0.17 |
| $r$ | −0.26 | −0.14 | −0.18 | 0.79 | 0.78 |



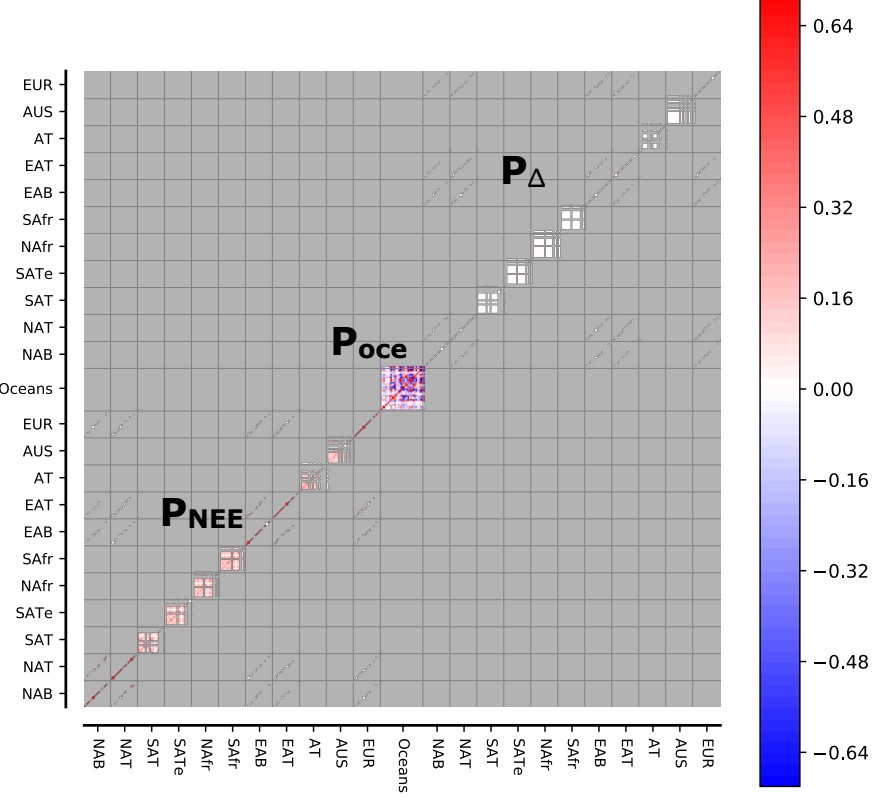

**Figure 3.** The prior **P** covariance structure represents squared uncertainty of the dimensionless state vector. The first $209 \times 209$ element block represents the covariance matrix for land NEE with a maximum diagonal uncertainty of 0.64 (equivalent to 80 %), the second $30 \times 30$ element block represents the covariance matrix for ocean fluxes with a maximum diagonal uncertainty of 1.0 (equivalent to 100 %), and the third $209 \times 209$ element block represents the covariance matrix for $\Delta_{\mathrm{ph}}$ with a maximum diagonal uncertainty of 0.04 (equivalent to 20 %). The matrix is organized according to TransCom ocean basins and land regions, where each land region contains 19 potential ecoregions (see Figs. 1 and 2).





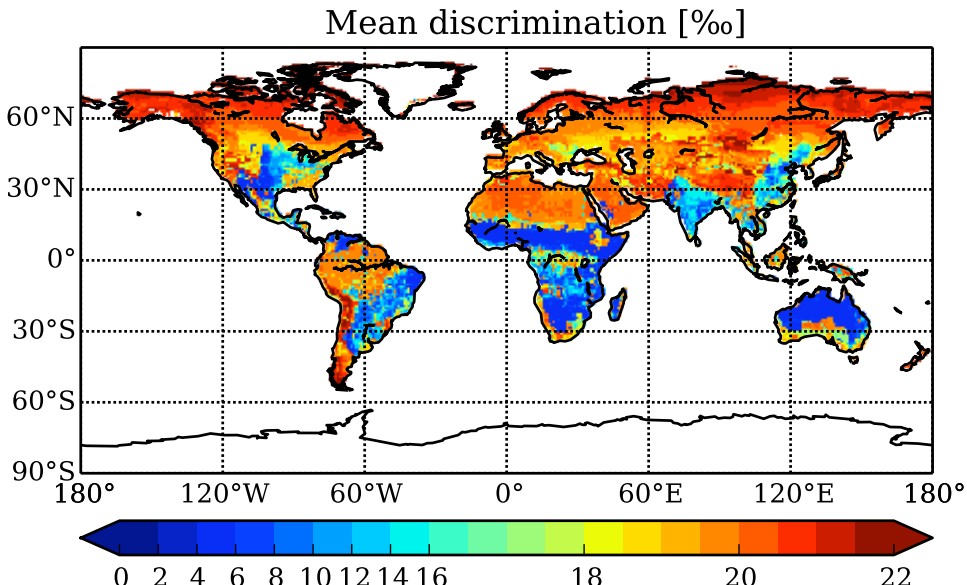

**Figure 4.** Mean (2001-2011) modeled discrimination parameter $\Delta_{ph}$ (‰) from SiBCASA. The discrimination is more detailed for $\Delta_{ph} >$ 16‰ to highlight the more subtle variations in $\Delta_{ph}$ in the dominant $C_3$ regions that experience different environmental forcing.





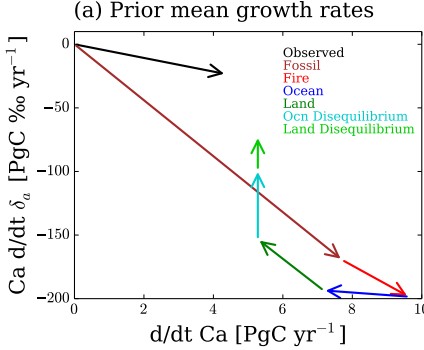
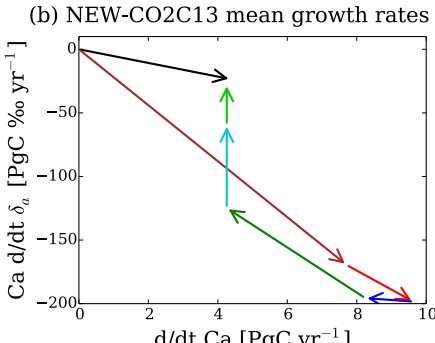

**Figure 5.** Annual mean carbon ($x$-axis) and $\delta^{13}$C ($y$-axis) growth rates for (a) the prior estimates and for (b) the NEW$-$CO2C13 experiment. Colored arrows represent the different sources and sinks of the carbon cycle. A closed budget for both tracers was accomplished in the NEW$-$CO2C13 experiment, as indicated by the resultant vector (sum of all colored arrows) returning to the black arrow (observed growth rate in atmosphere). To close the long-term trend we increased the isodisequilibrium fluxes by 20 %.





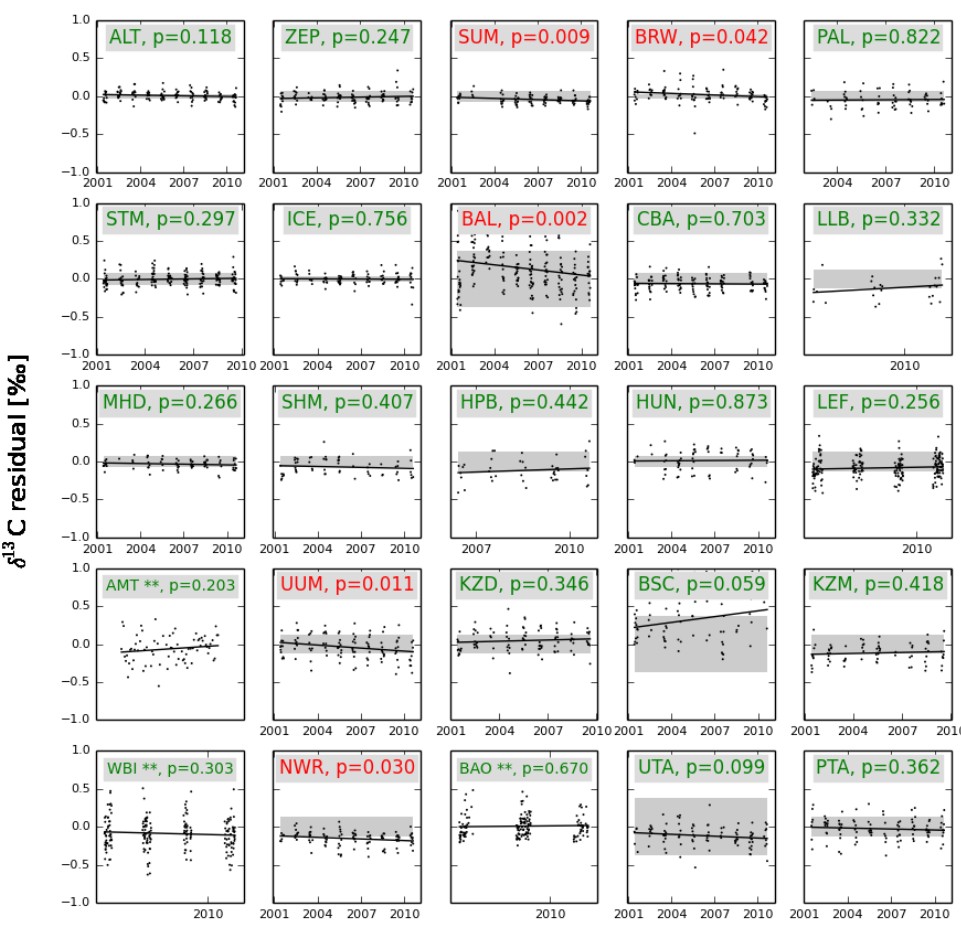

figure continues on the next page

**Figure 6.** Summer (JJA) residuals of $\delta^{13}$C [‰] in $CO_2$ for 46 sites (excluding aircraft and ships) situated in the Northern hemisphere. These sites are ordered based on the their latitudinal location; most Northern site is placed at the top left (Alert, Canada) and the site nearest to the equator at the bottom left on the next page (Christmas Island, Republic of Kiribati). All residuals (simulated minus observed) are calculated from a traditional $\mathrm{TRAD-CO2}$ inversion with scaled disequilibrium fluxes. Assuming a closed long-term mean budget in $\delta^{13}$C we tested the Ho hypothesis the slope of the linear regression line is zero. Sites with a trend where the p-value is smaller than the significance level of 5 % are shown in red, whereas the remaining sites without significant trend are shown in green. The sample uncertainty (model-data mismatch) used for the $\mathrm{NEW-CO2C13}$ and $\mathrm{NEW-2STEP}$ inversions is displayed by transparent gray areas. Sites marked with ** were not included in the inversions but were used for independent verification. For detailed information of the sites and their location we refer to the NOAA website: http://www.esrl.noaa.gov/gmd/ccgg/carbontracker/observations.php.





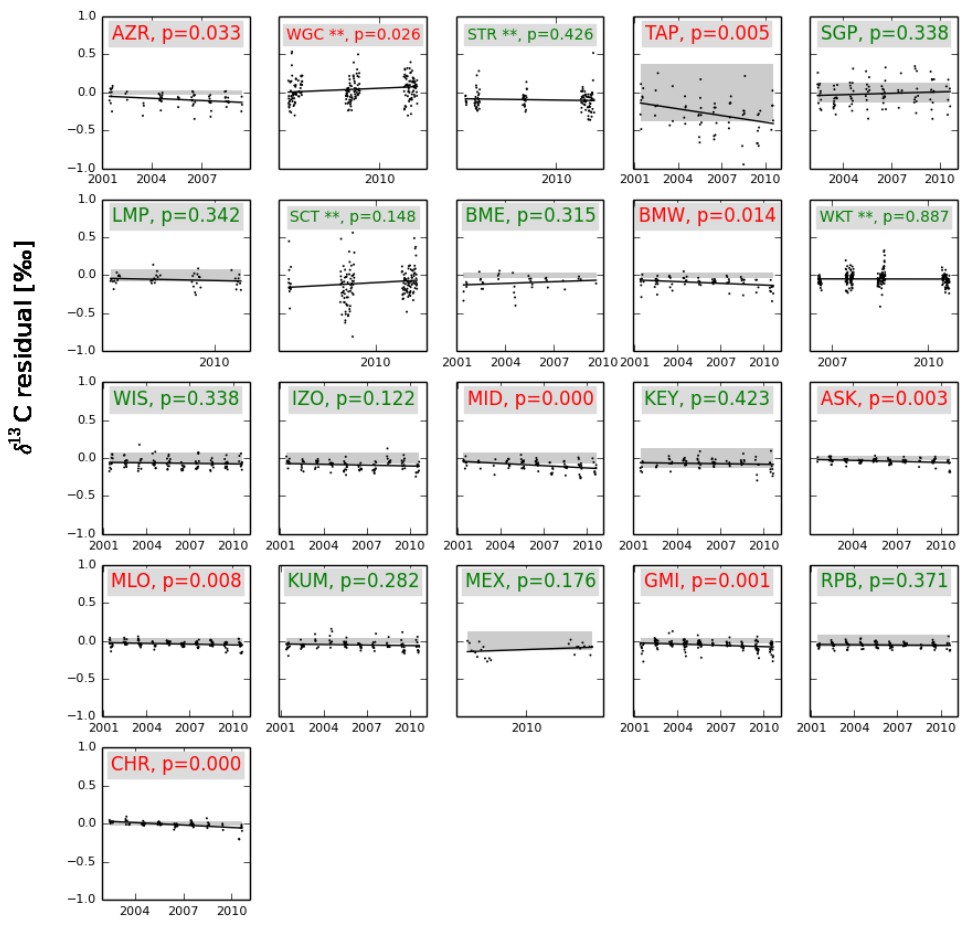





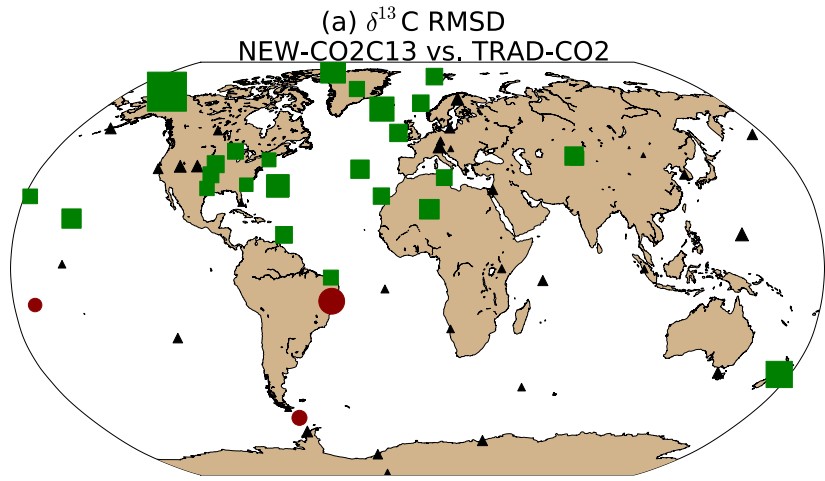

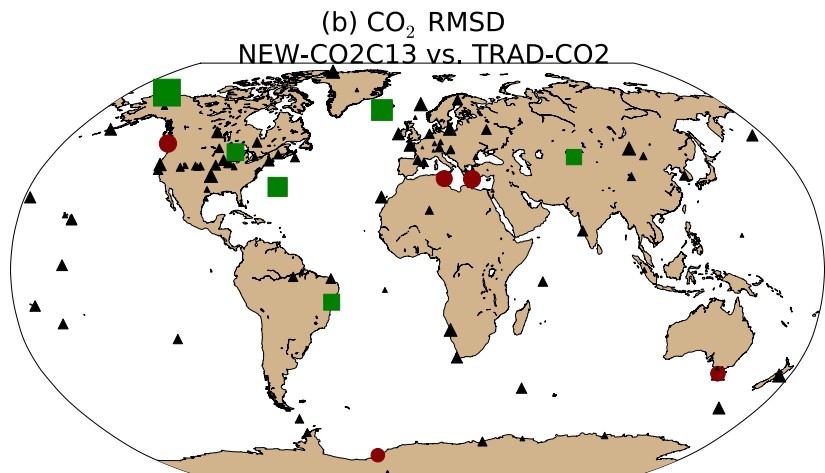

**Figure 7.** A comparison of the relative performance of inversion techniques for the period 2001 through 2006 based on the ratio of the model-data (a) $\delta^{13}C$ Root-Mean-Square-Difference (RMSD) of $NEW-CO2C13$ to $\delta^{13}C$ RMSD of $TRAD-CO2$, and (b) $CO_2$ RMSD of $NEW-CO2C13$ to $CO_2$ RMSD of the $TRAD-CO2$ inversion. A ratio lower than 1.0 indicates a higher accuracy of the $NEW-CO2C13$ inversion technique: green sites indicate a ratio $\leq 0.95$, red sites indicate a ratio $\geq 1.05$, and sites where the difference in respective RMSD's is less than 0.05 are given in black. The size of the each symbol is a measure of the relative performance of $NEW-CO2C13$ in comparison to $TRAD-CO2$. The larger the symbol, the more the ratio of RMSDs differs from 1.0.





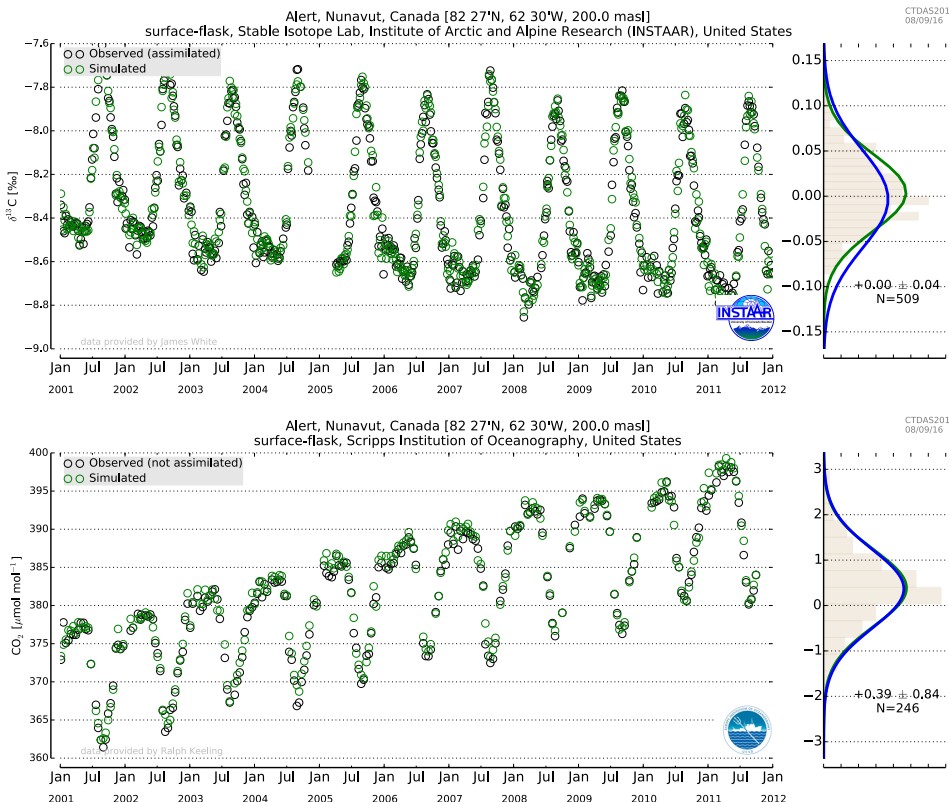

**Figure 8.** Comparison of two different inversion experiments at Alert (ALT, Canada). The top panel displays $\delta^{13}$C observations (black circles) together with simulated $\delta^{13}$C from NEW−CO2C13 (green circles). The top right panel displays the probability density functions (PDF) of the residuals between NEW−CO2C13 and observed (green) and between TRAD−CO2 and observed (blue). The lower panel displays independent flask measurements (not used in the assimilation) of $CO_2$ (black circles) at Alert with simulated $CO_2$ from NEW−CO2C13 (green circles). Notice the almost identical distribution of the residual PDFs between NEW−CO2C13 and TRAD−CO2 inversion techniques.





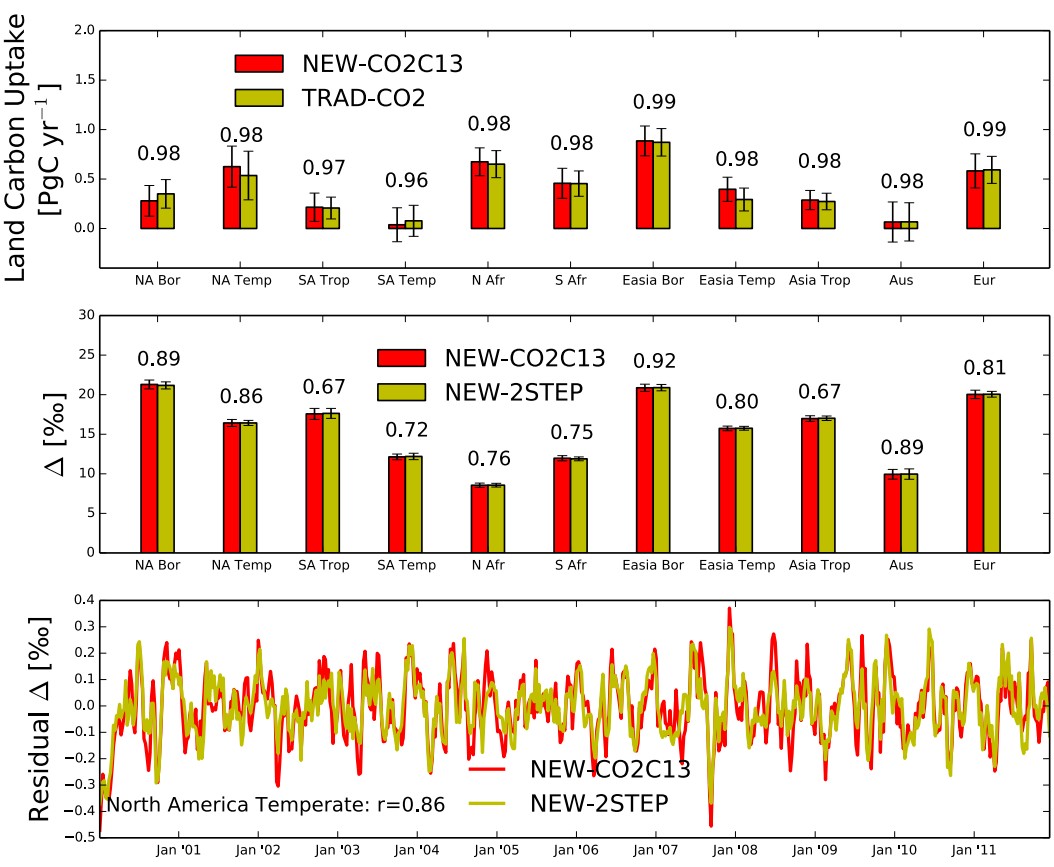

**Figure 9.** Top panel: the 11-year mean land carbon uptake $[\mathrm{Pg\,C\,yr^{-1}}]$ for each TransCom region with estimates from the nonlinear $\mathrm{NEW-CO2C13}$ inversion (red) and estimates from the linear $\mathrm{TRAD-CO2}$ inversion (yellow). Error bars depict $1\sigma$ standard deviation of the flux IAV. The 11-year correlation coefficients $r$ between the two inversion methods are given on top of the bars. These correlations are based on the 3-month boxcar mean anomalies after subtracting the seasonal cycle. Middle panel: comparison of $\Delta_{\mathrm{ph}}$ [‰] between the $\mathrm{NEW-CO2C13}$ inversion and the linear $\mathrm{NEW-2STEP}$ inversion. We again provide IAV error bars and correlation coefficients between inversion methods. Lower panel: the 3-month box car mean anomalies in $\Delta_{\mathrm{ph}}$ for the North America Temperate TransCom region to illustrate the high degree of similarity between both inversion methods ($r = 0.86$).



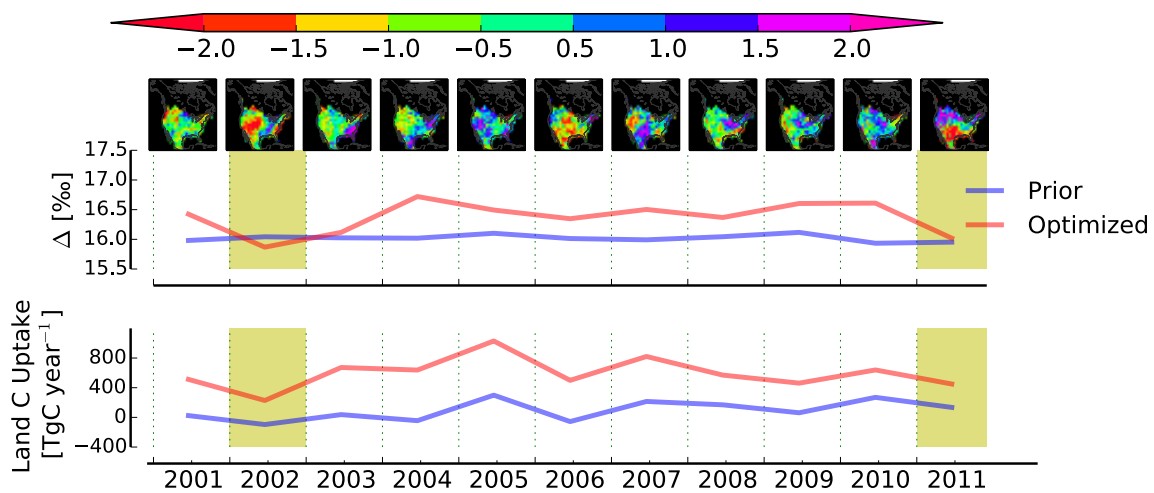

**Figure 10.** Top panels: the annual averaged Standardized Precipitation and Evaporation Index (SPEI) estimated for the North American Temperate domain (map inserts). Middle panel: the annual GPP weighted averaged $\Delta_{ph}$ [‰] of vegetation against $^{13}CO_2$ from NEW$-$CO2C13 (red) and SiBCASA (blue) estimated for the same domain. It illustrates the summertime isoforcing of $\delta^{13}C$ towards the atmosphere (as wintertime $\Delta_{ph}$ has no impact on atmospheric $\delta^{13}C$). Lower panel: net carbon uptake [TgC yr$^{-1}$] from NEW$-$CO2C13 (red) and SiBCASA (blue) estimated for the same domain. The yellow shaded years (2002 and 2011) indicate significant drought conditions as recorded in SPEI and other independent reports (e.g. *Seager*, 2010; *Schwalm et al.*, 2012; *Long et al.*, 2013). These droughts correlate with reductions in annual mean $\Delta_{ph}$, and reductions in the estimated carbon sinks as reported in *Peters et al.* (2007).





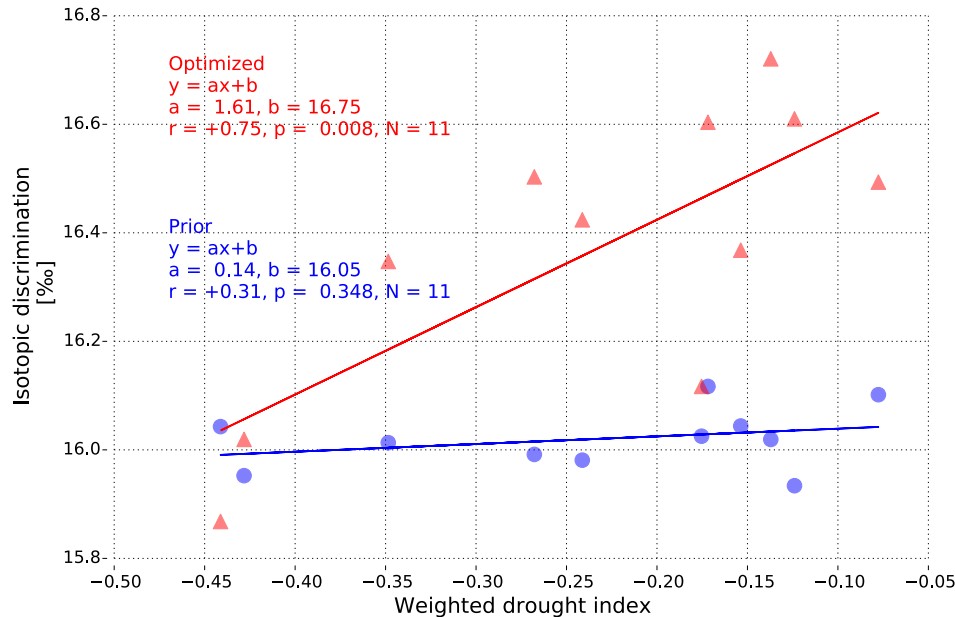

**Figure 11.** Weighted SPEI drought index (WDI) versus annual mean isotopic discrimination $\Delta_{\mathrm{ph}}$ integrated over North American Temperate domain. Results from the SiBCASA biosphere model (blue circles) show no significant correlation between $\Delta_{\mathrm{ph}}$ and large scale droughts, while the simultaneous optimization of carbon sinks and $\Delta_{\mathrm{ph}}$ with atmospheric $CO_2$ and $\delta^{13}C$ observations (red triangles) suggests a highly significant correlation can be derived. The slope of the red regression line is 1.61‰/WDI (95% confidence interval of a two-sided distribution with 9 degrees of freedom, p=0.008). The SiBCASA slope is however not significantly different from zero (p»0.05). The integrated $\Delta_{\mathrm{ph}}$ values are GPP-weighted per grid box as in Fig. 10. WDI is based on the SPEI index but area weighted to give years with large serried areas that experienced severe droughts (with SPEI smaller than -1.2) more leverage.