# Peer review of "The CarbonTracker Data Assimilation System for $CO_2$ and $\delta^{13}C$ (CTDAS-C13 v1.0): retrieving information on land-atmosphere exchange processes"

_Geoscientific Model Development, 2017_

## Short Comment (SC1) · 8 Jun 2017

Dear authors,

in my role as Executive editor of GMD, I would like to bring to your attention our Editorial version 1.1:

http://www.geosci-model-dev.net/8/3487/2015/gmd-8-3487-2015.html

This highlights some requirements of papers published in GMD, which is also available on the GMD website in the 'Manuscript Types' section:

http://www.geoscientific-model-development.net/submission/manuscript_types.html

[Figure]

In particular, please note that for your paper, the following requirement has not been met in the Discussions paper:

- "The main paper must give the model name and version number (or other unique identifier) in the title."

Please add the acronym CTDAS plus a unique version identifier of the exact version of the in this article described assimilation system in the title of your article in your revised submission to GMD.

Yours,

Astrid Kerkweg

––––––––––––––––––––––––––––––––––

---

## Referee Comment (RC1) · Anonymous Referee #1 · 30 Jun 2017

General comment: I read this manuscript with much interest. The authors did a good job in using atmospheric 13C data in a global data assimilation system for its additional information in partitioning land and ocean fluxes. Four assimilation experiments are logically designed to demonstrate the improvements in the global carbon cycle estimation brought by the use of the 13C data. The finding of the impact of drought on 13C discrimination is also interesting and convincing to a large extent. The manuscript is well written. However, the following issues need to be addressed before its publication.
1. Figures 10 and 11 show large (>0.5 permil) changes in the optimized plant 13C discrimination rate from the prior value, indicating that Eq. 2 for estimating the prior values does not perform well at least under drought conditions. The equation is based

on Suits et al. (2005) and includes full discrimination processes from free air to the photosynthetic site inside chloroplasts. However, there are various ways to implement the equation. It is not clear how Ci and Cc in the equation are estimated. Usually, stomatal conductance and mesophyll conductance are used to estimate them. In previous research, mesophyll conductance is often simply scaled to stomatal conductance. Chen et al. (2017, GMD) used a mesophyll model of Harley et al. (1992, Plant Physiology), and found it to be effective in improving the sensitivity of the modeled 13C discrimination rate to environmental conditions and in removing abnormal values caused by scaling mesophyll conductance to stomatal conductance. I am not requesting the authors to further develop their prior model for this paper, but they should make it clear how the equation is implemented and discuss issues associated with photosynthetic discrimination modeling. Perhaps they should also estimate the errors in their modeled discrimination rate. These errors would have implication on the partition between land and ocean fluxes, seasonal variability of the fluxes and the drought effect found in the manuscript. 2. I appreciate very much that both land and ocean discrimination rates are optimized in their data assimilation systems, and it is interesting to see that it is possible that these rates can be optimized with currently available measurements. The authors also make it clear that these optimizations are based on the assumption that the prior disequilibrium fluxes of land and ocean have no bias errors. We understand that these disequilibrium fluxes are large and nearly equivalent to discrimination fluxes in size and that their estimates are quite involved and inaccurate. I wonder what is the justification to optimize discrimination but not disequilibrium. Since the disequilibrium rates over both land and ocean are difficult to estimate accurately, I wonder what are the impacts of their errors on the optimized fluxes and discrimination rates. The authors qualitatively discussed these impacts in Discussion, but the discussion is not useful for assessing the reliability of optimized results of their data assimilation systems. It would be useful to do a quantitative assessment of these impacts. 3. The word "multi-species" in the title is a bit misleading because there are only two gas species, CO2 and 13CO2, considered in their data assimilation systems, while multi-species would imply at least

three species. Although the systems are intended for more than two species, the current study only uses two species. I suggest changing it to duel-species or some other phrases.

---

## Referee Comment (RC2) · P. Rayner (Referee) · 16 Jul 2017

This paper presents an extension to the CarbonTracker system to ingest measurements of the C13 isotopic signal in atmospheric CO2 samples. It demonstrates the simultaneous optimisation of photosynthetic fractionation and net CO2 exchange. This makes it a significant extension over previous uses of C13 in atmospheric inversions so it is both within scope for GMD and significance.

Like many of us who have battled with inversions using C13, the authors have made some choices about what can and cannot be inferred from this species given the associated nuisance variables of isotopic disequilibrium flux (isoflux). They pretty much

remove the possibility of C13 observations informing long-term mean fluxes by closing the C13 budget with a tuning of the isoflux. they are careful to keep this in mind and refrain from commenting on the long-term mean fluxes.

This is quite a useful contribution to the reemergence of C13 as a constraint, especially addressing the weakness of fixed fractionation of many previous studies. I have two concerns I would like the authors to address.

The first is a bit more detail on posterior uncertainties. This is more difficult in the NKF formalism of CarbonTracker than for the classical synthesis inversion but, especially in the nonlinear case, some sense of ensemble correlations among fractionations and fluxes would be useful. Perhaps these are the correlations already quoted, it seemed from the text these were signal correlations. As a side-note, the p-values attached to the correlations are not relevant here. We are interested in the strength of a relationship while the p-value shows the chance of giving such a correlation if the population value was zero.

My second concern is raised by the authors in the discussion but is not really dealt with. It could affect some of the conclusions. The authors note (P17) that impacts of changing net flux or fractionation on the isoflux are neglected. they correctly diagnose that the problem arises because the isoflux is not included in the optimisation. they suggest one solution, the partition of net flux into its gross components. There is another approximate solution. The main result of this process is a dilution of C13 signals by the isoflux. This can be parameterised as a response function for the C13 signal from a net flux. this was how Rayner et al., 1999 approached the problem, taking response functions from Trudinger et al. 1999. The time-scales for this response are long cf the assimilation window used in CarbonTracker so I'm not sure whether one can even capture the effect but we did find it had an impact on interannual variability. The problem may be less severe for the current paper because the prior signal for this response should be captured by SiBCASA. To quantify the effect I recommend that the authors take the difference between their prior and posterior flux and transport its C13

signature with and without the dilution response. This should at least give a sense of the significance of the problem.

---

## Author Comment (AC1) · 23 Sep 2017

**Replies to the Editor and Reviewers**

**MS No.: gmd-2017-84**

**September 21 2017**

We thank the Executive Editor and the Reviewers for their helpful and supportive comments. Their thorough analysis, critical comments, and suggestions have helped us to improve and sharpen the manuscript.

**Dear Astrid Kerkweg, Executive Editor,**

General comment:

*Please add the acronym CTDAS plus a unique version identifier of the exact version of the in this article described assimilation system in the title of your article in your revised submission to GMD.*

**Authors:** We changed the title of the manuscript to: The CarbonTracker Data Assimilation System for $CO_2$ and $\delta^{13}C$ (CTDAS-C13 v1.0): retrieving information on land-atmosphere exchange processes

**Dear Reviewer #1,**

General comments:

*Figures 10 and 11 show large (>0.5 permil) changes in the optimized plant 13C discrimination rate from the prior value, indicating that Eq. 2 for estimating the prior values does not perform well at least under drought conditions. The equation is based on Suits et al. (2005) and includes full discrimination processes from free air to the photosynthetic site inside chloroplasts. However, there are various ways to implement the equation. It is not clear how Ci and Cc in the equation are estimated. Usually, stomatal conductance and mesophyll conductance are used to estimate them. In previous research, mesophyll conductance is often simply scaled to stomatal conductance.*

**Authors:** We thank the Reviewer for making us aware that some parts of the SiBCASA's model description were unclear. We improved its description from page 8 line 24 through page 9 line 8 in the new manuscript. However, we choose not to provide all the equations that are involved in the photosynthesis calculations. The main focus of this manuscript is to describe our first dual-species inverse modeling framework of CTDAS. Therefore, a detailed description of the prior flux estimates is less relevant here. Instead, we give the necessary references to the important papers that describe them in detail.

*Chen et al. (2017, GMD) used a mesophyll model of Harley et al. (1992, Plant Physiology), and found it to be effective in improving the sensitivity of the modeled 13C discrimination rate to environmental conditions and in removing abnormal values caused by scaling mesophyll conductance to stomatal conductance. I am not requesting the authors to further develop their prior model for this paper, but they should make it clear how the equation is implemented and discuss issues associated with photosynthetic discrimination modeling.*

**Authors:** The $CO_2$ concentration close to the chloroplasts ($C_c$) is derived with the mesophyll conductance ($g_m$) estimate ($C_c = C_i - A_n/g_m$). Mesophyll conductance inside SiBCASA is not scaled to stomatal conductance but is derived with the following function (Suits et al. 2005):

$$g_m = 4000 \cdot vmax0 \cdot \Pi \cdot \beta,$$

where 4000 is a constant used to achieve a drop of $CO_2$ partial pressure of 8 Pa between $C_i$ and $C_c$ when assimilation rate is high, vmax0 is the maximum potential photosynthetic rate at the top of canopy, $\Pi$ is a factor that expresses the integrated photosynthetic rate over the entire canopy and ß is the soil moisture stress parameter, which is a function of the plant available water in the soils. That means water stress limits not only the assimilation rate by scaling down Vmax but also increases the mesophyll resistance, making it harder for $CO_2$ molecules to diffuse to the leaf chloroplasts. This gives a $CO_2$ ratio $C_c/C_a$ that scales with the carboxylation discrimination parameter ($\Delta_f$) used to predict the total discrimination (Eq. 2 in manuscript). Neglecting mesophyll conductance would mean $C_c$ equals $C_i$ even though $C_c$ can be significantly lower. This would require an often-used simplified version of the discrimination model where $\Delta_f$ scales linearly with $C_i/C_a$, and therefore risking an overestimation of discrimination. A previous comparison between observed and simulated $\delta^{13}C$ demonstrated the ability of SiBCASA to fit the data within 1 ‰ for a selection of sites of the BASIN network (van der Velde et al., 2014).

We make reference to the mesophyll conductance formulation by Suits et al. (2005) on page 8 line 31. We also make the importance of mesophyll conductance in relation to photosynthesis and discrimination more explicitly clear in the manuscript (page 9 line 1). The implementation of alternative mesophyll conductance models is certainly something we want to explore in the future. However, we did not find abnormal large or small values in discrimination caused by mesophyll conductance as claimed by Chen et al. (2017). On the contrary, the current photosynthesis model in SiBCASA conserves discrimination to an extreme due to lack of soil moisture sensitivity. But once again, a detailed discussion of the prior flux/discrimination uncertainties is out of scope of this manuscript and is either already done elsewhere (e.g. Suits et al. 2005, van der Velde et al. 2013, van der Velde et al. 2014) or is currently in preparation as a separate manuscript.

*I appreciate very much that both land and ocean discrimination rates are optimized in their data assimilation systems, and it is interesting to see that it is possible that these rates can be optimized with currently available measurements.*

**Authors:** We want to emphasize that ocean discrimination is not optimized. Only terrestrial discrimination, and the net ocean and land carbon exchange fluxes. The main reason is that ocean discrimination is much smaller in magnitude than terrestrial discrimination, and spatiotemporally much more homogeneous (page 5 line 17). This is furthermore mentioned on page 6 line 15, and page 10 line 11.

*The authors also make it clear that these optimizations are based on the assumption that the prior disequilibrium fluxes of land and ocean have no bias errors. We understand that these disequilibrium fluxes are large and nearly equivalent to discrimination fluxes in size and that their estimates are quite involved and inaccurate. I wonder what is the justification to optimize discrimination but not disequilibrium. Since the disequilibrium rates over both land and ocean are difficult to estimate accurately, I wonder what are the impacts of their errors on the optimized fluxes and discrimination rates. The authors qualitatively discussed these impacts in Discussion, but the discussion is not useful for assessing the reliability of optimized results of their data assimilation systems. It would be useful to do a quantitative assessment of these impacts.*

**Authors:** Although we did not jointly optimize disequilibrium inside CTDAS-C13, we optimized it offline to match the 11-year trend in observed $\delta^{13}$C. We found that the prior disequilibrium estimates underestimated the $\delta^{13}$C trend substantially by 20 %. These bottom-up estimates were scaled by a factor of 1.2 for use in CTDAS-C13. This gave us an acceptable 11-year mean fit for most of the $\delta^{13}$C sites in the Northern and Southern Hemispheres with a RMSD of 0.079 ‰ and the mean bias of -0.01 ‰ (at $\delta^{13}$C measurement precision). This procedure is described in the manuscript on page 12, line 18. We believe however that the choice of 1.2 will not have a significant impact on our conclusions, because we focused only on interannual variability in fluxes and discrimination. The scaling assured that land and ocean $CO_2$ flux magnitudes remained close to the results of the traditional $CO_2$-only inversion.

The extent disequilibrium fluxes will change year to year is a matter of debate in the literature. It has been suggested that at least the gross flux component of the disequilibrium between the atmosphere and ocean or terrestrial biosphere are mostly controlled by large-scale thermodynamic influences and are expected to change little on interannual time scales (e.g. Rayner et al. 1999), but other studies suggest this is not necessarily be true (e.g. Alden et al. 2010). Given the short assimilation window of 5 weeks that is used in CTDAS-C13 it is doubtful whether we can capture such effects that are dampened by their large reservoir size.

To quantify the influence of errors in the disequilibrium flux we performed an additional experiment with SiBCASA where we allowed extra variability in respiration and discrimination to drive through the disequilibrium isoflux. For a detailed explanation of this experiment you can read our reply to Reviewer #2 (Prof. Rayner). We would indeed generate extra variability in the terrestrial disequilibrium budget term, necessitating 10 % less variability in discrimination to keep the $\delta^{13}$C budget closed. It indicates that allowing for errors in the disequilibrium fluxes the variations in discrimination could in reality be slightly larger or smaller than estimated in the paper, but are nonetheless still much larger than in SiBCASA (twice as large standard deviation). We make reference to this experiment in the manuscript on page 18, line 1.

*The word "multi-species" in the title is a bit misleading because there are only two gas species, CO2 and 13CO2, considered in their data assimilation systems, while multi-species would imply at least three species. Although the systems are intended for more than two species, the cur- rent study only uses two species. I suggest changing it to duel-species or some other phrases.*

**Authors:** We changed the title of the manuscript to *The CarbonTracker Data Assimilation System for $CO_2$ and $\delta^{13}$C (CTDAS-C13 v1.0): retrieving information on land-atmosphere exchange processes*. Other references to *multi-species* in the main text are changed to *dual-species*.

**Dear Prof. Rayner, Reviewer #2,**

1st concern:

*The first is a bit more detail on posterior uncertainties. This is more difficult in the NKF formalism of CarbonTracker than for the classical synthesis inversion but, especially in the nonlinear case, some sense of ensemble correlations among fractionations and fluxes would be useful. Perhaps these are the correlations already quoted, it seemed from the text these were signal correlations.*

**Authors**: Interpreting the ensemble correlations between $\Delta$ (fractionation) and fluxes is indeed difficult with CTDAS, specifically because we cannot calculate covariances structures over scales beyond the current assimilation window of 5 weeks. So our short time window prevents us from deriving reliable seasonal or annual mean uncertainties and correlations from its covariance matrix. Within the 5-week windows, we can look at these covariations though and as we show here below in Figure 1, the monthly posterior ensemble-correlations ($N=150$) between NEE and $\Delta$ aggregated for TransCom regions are, besides the diagonal, just small (between -0.3 and 0.3). However, between the linear 2-step inversion and the nonlinear inversion the posterior ensemble-correlations are markedly similar, which suggest a shared commonality of the internal error estimate between the two methods. In the manuscript however, we show the high degree of similarity of posterior $\Delta$ between the two inversion methods by calculating their temporal correlations over a 11-year period (Figure 9 in the manuscript). Table 3, as the Reviewer had deduced, indeed presents a signal correlation rather than a spatiotemporal covariance.

As argued by Peters et al. (2005), the formal uncertainty estimate cannot be based on ensemble-correlations but should instead be based on a number of different inversions, with different assumptions and model setups. As mentioned in the manuscript (page 17, line 21) the reported correlations between NEE and $\Delta$, and the drought index and $\Delta$ remain robust and significant if we change the atmospheric transport characteristics, the optimization method (linear vs. nonlinear), or the assigned model-data error in CTDAS. Not only robust for the Northern Hemisphere but also for smaller subregions like Europe and parts of Eurasia. These additional experiments and results are in preparation for a second publication and demonstrate in more detail atmospheric $\delta^{13}C$ as a new observational constraint of the impact of droughts on the water-use efficiency using CTDAS-C13.

*As a side-note, the p-values attached to the correlations are not relevant here. We are interested in the strength of a relationship while the p-value shows the chance of giving such a correlation if the population value was zero.*

**Authors:** The p-values in the main text of the manuscript refer to the hypothesis test for slopes; can we reject the null hypothesis that the slope parameter is zero, i.e., is there a significant slope between NEE (or drought index) and $\Delta$ over a 11-year period? All slopes of the curve fits in the manuscript were tested using two-tailed distributions and a 95% confidence level for N-2 degrees of freedom. We improved the description in the main text.

[Figure]

Figure 1: Posterior NEE and Δ ensemble-correlation matrix for July 2002 where we aggregated the ecoregions to 11 TransCom regions (see Figure 2 in manuscript). The matrix on the left is derived from the linear 2-step inversion and the matrix on the right from the nonlinear inversion. In each matrix, the first 11x11 element block contains the NEE correlations between the TransCom regions, and the second 11x11 element block along the diagonal contains Δ correlations between the TransCom regions. The two off-diagonal 11x11 element blocks contain the correlations between NEE and Δ. In both matrices the diagonal correlations are 1.0, but the color scale limits values between -0.5 and 0.5.

2nd concern:

*My second concern is raised by the authors in the discussion but is not really dealt with. It could affect some of the conclusions. The authors note (P17) that impacts of changing net flux or fractionation on the isoflux are neglected. they correctly diagnose that the problem arises because the isoflux is not included in the optimisation. they suggest one solution, the partition of net flux into its gross components. There is another approximate solution. The main result of this process is a dilution of C13 signals by the isoflux. This can be parameterised as a response function for the C13 signal from a net flux. this was how Rayner et al., 1999 approached the problem, taking response functions from Trudinger et al. 1999. The time-scales for this response are long cf the assimilation window used in CarbonTracker so I'm not sure whether one can even capture the effect but we did find it had an impact on interannual variability. The problem may be less severe for the current paper because the prior signal for this response should be captured by SiBCASA. To quantify the effect I recommend that the authors take the difference between their prior and posterior flux and transport its C13 signature with and without the dilution response. This should at least give a sense of the significance of the problem.*

**Authors**: Prof. Rayner makes a valid point that our current optimization framework lacks C13 signal dilution that could result in an over- or underestimation of the Δ variability. The Δ variability would in reality drive through the disequilibrium isoflux and generate additional variability, but this was ignored in our framework. We found by analyzing the variability in the net flux and isoflux components of the $CO_2 * \delta 13C$ mass balance that such 'dilution effects' of the C13 signals were in fact quite minimal. Variability in Δ with dilution response was about 10% smaller in comparison to the Δ estimate without dilution response. However, both solutions still contained significantly more (two times more in standard deviation) variability in Δ than predicted by SiBCASA.

To demonstrate this, consider the following formulation for rate of change of atmospheric $\delta^{13}C$ only due to terrestrial C13 exchange:

$$\frac{dC_a\delta_a}{dt} = N_b\delta_{ab} + F_{ba}(\delta_{ba} - \delta_{ab})$$

The first term on the right-hand side represents a net isoflux: the product of net terrestrial carbon uptake ($N_b$) and isotopic signature in carbon assimilation ($\delta_{ab}$), where $\delta_{ab}$ is the sum of $\delta_a$ (atmospheric $\delta^{13}C$) and $\Delta$ (photosynthetic fractionation). The second term represents the disequilibrium isoflux: the product of total respiration ($F_{ba}$) and the difference in isotopic signatures of the total respiration and photosynthesis ($\delta_{ba} - \delta_{ab}$). To account for dilution effect, extra $\Delta$ variability should go into $\delta_{ab}$ and $\delta_{ba}$. There is an immediate effect in $\delta_{ab}$ because $\delta_{ab} \approx \delta_a - \Delta$. For $\delta_{ba}$ the response time is more complex, because the $\Delta$ variations return quickly via autotrophic respiration (perhaps ~1-7 days), while there is a much more dampened effect of $\Delta$ variations in the heterotrophic component of $\delta_{ba}$ because the multiple carbon pools with different turnover rates homogenize the isotopic signature of carbon released from the soils (Alden et al., 2010). For this analysis, we took monthly 1x1 degree $N_b$, $F_{ba}$, $\delta_{ab}$ and $\delta_{ba}$ from SiBCASA for the North American domain over a total time period of 11 years. We introduced for each year a spatial uncertainty parameter $\beta$ of +/-1‰ ($1\sigma$ standard deviation) and a covariance length scale of 300 km to mimic increased correlated interannual variability in $\delta_{ab}$ and $\delta_{ba}$. Because 40% of the North American respiration is heterotrophic (according to SiBCASA), and rather insensitive to changes in $\Delta$, we scaled down the $\beta$ parameter applied on $\delta_{ba}$ by a factor of 0.6. A second uncertainty parameter $\gamma$ was introduced to mimic 20% more variability in $F_{ba}$ using a covariance length scale of 300 km.

We rewrite the rate of change equations for two different cases: (1) with dilution response and (2) without dilution response, which is similar to the scenario portrait in the manuscript.

$$\frac{dC_a\delta_a}{dt}_{\text{with dilution}} = N_b(\delta_{ab} + \beta) + (F_{ba} + \gamma) \cdot [(\delta_{ba} + \beta \cdot 0.6) - (\delta_{ab} + \beta)] \qquad (1)$$

$$\frac{dC_a\delta_a}{dt}_{\text{without dilution}} = N_b(\delta_{ab} + \beta) + F_{ba}(\delta_{ba} - \delta_{ab}) \qquad (2)$$

How much variability in $\delta_{ab}$ in eq. (2) do we need to match the more realistic 'with dilution' scenario? Answer can be deduced after some substitution between eq. (1) and eq. (2):

$$\hat{\delta}_{ab} = \frac{\frac{dC_a\delta_a}{dt}_{\text{with dilution}} - F_{ba}(\delta_{ba} - \delta_{ab})}{N_b}$$

$\hat{\delta}_{ab}$ estimation is analogous with the $\Delta$ estimation in the manuscript using atmospheric constraints and a fixed terrestrial disequilibrium flux without dilution effects. Figure 2 below shows that variability of the North American annual mean $\hat{\delta}_{ab}$ is overestimated (10% larger standard deviation) in comparison to the 'correct' estimate of ($\delta_{ab} + \beta$). More importantly, variations in both ($\delta_{ab} + \beta$) and $\hat{\delta}_{ab}$ are significantly larger than SiBCASA's estimate for $\delta_{ab}$ (two times larger standard deviation) and highly correlated (r=0.97). It suggests the possibility that variations in $\Delta$ may be overestimated in the manuscript, however, the dilution response introduced via the gross fluxes and isotopic signatures is not large enough to believe our current $\Delta$ estimates from CTDAS-C13 are unrealistic. We make a reference to this experiment on page 18, line 1.

[Figure]

Figure 2: North American mean estimates of $\delta_{ab}$ from SiBCASA (cyan), $\delta_{ab}$ with extra $\beta$ variability (blue), and the top-down estimate $\hat{\delta}_{ab}$ that excludes dilution effects (red). Both $(\delta_{ab} + \beta)$ and $\hat{\delta}_{ab}$ are significantly larger than SiBCASA's estimate for $\delta_{ab}$ (more than 100% larger standard deviation) and highly correlated (r=0.97).

**References:**

Alden, C. B., J. B. Miller, and J. W. C. White (2010), Can bottom-up ocean $CO_2$ fluxes be reconciled with atmospheric [13]C observations?, *Tellus B*, *62*(5), 369–388, doi:10.1111/j.1600-0889.2010.00481.x.

Chen, J. M., G. Mo, G., and F. Deng (2017), A joint global carbon inversion system using both $CO_2$ and [13]$CO_2$atmospheric concentration data, Geosci. Model Dev., 10, 1131-1156, https://doi.org/10.5194/gmd-10-1131-2017, 2017.

Peters, W., J. B. Miller, J. Whitaker, A. S. Denning, A. Hirsch, M. C. Krol, D. Zupanski, L. Bruhwiler, P. P. Tans (2005), An ensemble data assimilation system to estimate $CO_2$ surface fluxes from atmospheric trace gas observations, *Journal of Geophysical Research*, *110*, D24304, doi:10.1029/2005JD006157.

Rayner, P.J., I. G. Enting, R. J. Francey and R. Langenfelds (1999), Reconstructing the recent carbon cycle from atmospheric $CO_2$, d13C and $O_2/N_2$ observations, Tellus (1999), 51B, 213–232

Suits, N., A. Denning, J. Berry, and C. Still (2005), Simulation of carbon isotope discrimination of the terrestrial biosphere, *Global Bio- geochem. Cycles*, *19*, GB1017, doi:10.1029/2003GB002141.

Van der Velde, I. R., J. B. Miller K. Schaefer, K. A. Masarie, S. Denning, J. W. C. White, P. P. Tans, M. C. Krol, and W. Peters (2013), Biosphere model simulations of interannual variability in terrestrial 13C/12C exchange, *Global Biogeochem. Cycles*, *27*(3), 637–649.

Van der Velde, I. R., J. B. Miller K. Schaefer, G. R. van der Werf, M. C. Krol, and W. Peters (2014), Terrestrial cycling of 13CO2 by photosynthesis, respiration and biomass burning in SiBCASA, *Biogeosciences*, *11*, 6553–6571.